# Specific seismic retrofitting of a compact reinforced concrete building with X-bracings and steel jackets. Application to a primary school in Huelva

Emilio Romero-Sánchez[☯], Antonio Morales-Esteban[ID]*[☯], María-Victoria Requena-García-Cruz[☯], Beatriz Zapico-Blanco[ID][☯], Jaime de-Miguel-Rodríguez[☯]

Department of Building Structures and Geotechnical Engineering, University of Seville, Seville, Spain

☯ These authors contributed equally to this work.
* ame@us.es

**Data Availability Statement:** All relevant data are within the paper manuscript and its Supporting

## Abstract

Previous research has indicated that many European buildings are vulnerable to moderate-magnitude earthquakes. For example, during the L´Aquila (Italia, $M_w$ 6.3, 2009) and Lorca (Spain, $M_w$ 5.9, 2011) earthquakes, many old buildings were severely damaged and some of them collapsed. In specific, significant damage has been found in several school buildings after past earthquakes in Europe. This is due to the fact that many of them were constructed prior to the current seismic codes, thus considering only gravitational loads and with no seismic design whatsoever. Primary schools are even more vulnerable than other typologies because of their low adult/child ratio. The seismic activity of the Iberian Peninsula is low-moderate. However, the Algarve and Huelva regions, which are situated in the south-west, are influenced by large faults which have caused major earthquakes of long-return periods. The European project PERSISTAH (*Projetos de Escolas Resilientes aos SISmos no Território do Algarve e de Huelva*, in Portuguese*)* aims to cooperatively evaluate the seismic vulnerability of primary schools in the Algarve (Portugal) and Huelva (Spain) regions. The present work is framed under this project. The objective of this paper is to determine the most effective retrofitting scheme for a typical primary school building in this area, considering structural, architectural and constructive parameters. The scheme could be applied to several buildings of the same typology, decreasing costs and time. An existing reinforced concrete frame building has been selected for the study. This is one of the most commonly used typologies for primary schools in this area. A nonlinear static analysis has been carried out in order to study its seismic behaviour. The performance point of the building has been obtained through the capacity-demand spectrum method. The preliminary results have confirmed the poor seismic behaviour of this building. Specifically, soft-story behaviour has been identified in the ground floor and short columns have been observed in the upper floors. Therefore, specific seismic retrofitting solutions have been proposed and evaluated in order to identify the one that is the most efficient. The combination of reinforcements has been done considering the structural and architectural impact and constructive parameters. The calculations have shown that steel X-bracings are the best solution for preventing the

Information files. The original files where obtained from the "Municipal file of Huelva" (Archivo municpal de Huelva, Casa Colón, Huelva, Spain).

**Funding:** This work has been supported by the INTERREG-POCTEP Spain-Portugal programme and the European Regional Development Fund through the 0313_PERSISTAH_5_P project and the VI-PPI of the University of Seville by the granting of a scholarship. The grant provided by the Instituto Universitario de Arquitectura y Ciencias de la Construcción is acknowledged.

**Competing interests:** The authors have declared that no competing interests exist.

formation of a soft-storey in the ground floor. Unfortunately, this scheme increases the deformation in the upper floor columns. The best solution for the upper floors' short columns has been the use of steel jackets. The results have also shown that this combination produces an important reduction of the expected general damage level. The resulting retrofitting scheme can be extrapolated to other buildings with a similar typology.

## Introduction

The works described here are concentrated, in particular, in the Spanish province of Huelva. The seismic activity of the IP is low-moderate. However, the Algarve and Huelva regions, which are situated in the southwestern IP, are influenced by the contact area between the Eurasian and the African tectonic plates. This area is characterised by the presence of large faults, such as the Azores-Gibraltar faults [1], which produce large earthquakes (Mw≥6) of long-return periods [2] (Fig 1). The San Vicente Cape and Horseshoe faults are also relevant and caused the most outstanding earthquake that affected the IP. This was the 1755 Lisbon earthquake and tsunami ($M_w = 8.5$) [3], which is the largest documented seismic event that has affected Europe, killing up to 100,000 people. Another relevant earthquake produced in these faults was the 1969 earthquake ($M_w = 8$) [4]. These large earthquakes have long return periods, which makes the population unaware of the seismic hazard.

It is possible to reduce the seismic risk by improving the prevention studies and emergency plans. In this sense, it is important to analyse the seismic vulnerability of the buildings given that a large part of the losses is due to the deficient seismic behaviour of the building structures.

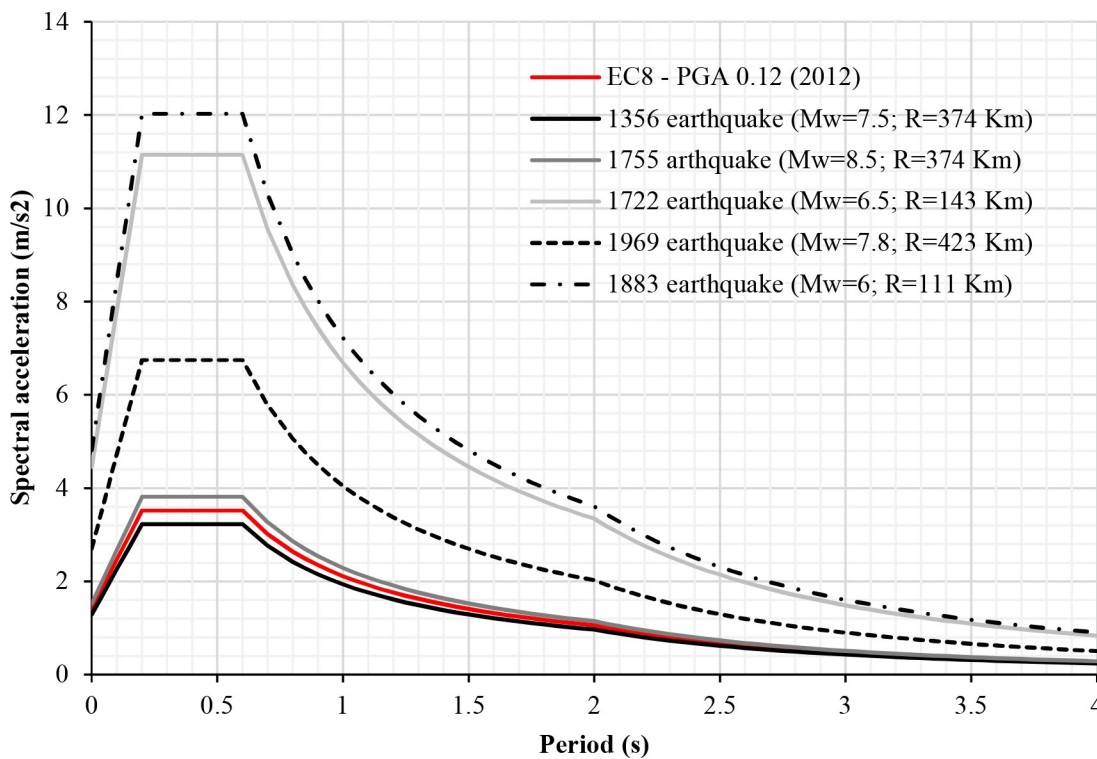

**Fig 1. Comparison of response spectrums of EC-8 and NCSE-02 with the response spectrums of historic earthquakes.**

Therefore, the study of seismic retrofitting techniques, which improves the buildings' seismic behaviour, is mandatory [5].

When assessing a community's vulnerability, the physical damage is not the only important factor; other indirect effects, such as economic losses, social disorder and lack of resilience of the population, must also be accounted for.

In this context, schools are especially relevant as they play a key role in our society. Schools are considered a security reference for children and, therefore, they are a cornerstone for the creation of resilient communities [6]. In addition, in the event of any natural disaster, schools must be a refuge for the population. For all of these reasons, in the case of an earthquake, school buildings must not be severely damaged. Finally, normality is considered to be recovered when schools resume their regular activity. Increasing the safety of primary schools also has an important impact on society because they are spread across the entire territory, as there is at least one school in each city or town.

School buildings located in the Huelva region have a high seismic vulnerability due to the following aspects: their low adult/child ratio and their configuration. The child population (kids between 3 to 11 years old) are the most vulnerable people in the society. The building´s configuration is characterized by the presence of several seismic weak points (soft storeys at ground floors, plan irregularities or short columns). Moreover, these buildings were constructed prior to the current seismic resistant codes and their structures were calculated considering only gravity loads. The buildings designed without considering the seismic action were seriously damaged in the Lorca earthquake [7]. Likewise, these buildings are simple and repetitive which allows us to extrapolate the results.

The works described in this paper are framed under the European project PERSISTAH (*Projetos de Escolas Resilientes aos SISmos no Território do Algarve e de Huelva*, in Portuguese), cooperatively developed between Spain and Portugal [8, 9]. The main goal of the project is to assess the seismic vulnerability of primary schools located in the Algarve (Portugal) and Huelva (Spain). If there is an earthquake, both regions will be equally affected. The project also intends to educate children, who are the future of our society, to create resilient communities.

The work's principal objective is to obtain the most efficient seismic retrofitting solution. In that sense, there are several paramount aspects which need to be considered. First, it is important for the retrofitting scheme to provide a better global seismic behaviour in the structure. Second, it must be integrated into the building, causing the least possible architectural impact. Third, the construction process must be as easy and as quick as possible to avoid any hindrance to the teaching activities. Finally, the retrofitting scheme needs to be applicable to other buildings with a similar typology (type and structural system).

For this research, a building has been selected with a typical structural typology, which is one of the most common in the study area. This is a Reinforced Concrete (RC) frame building which has been widely used in primary schools. Due to that, the seismic retrofitting results have a greater impact and it is more easy to extrapolate to other buildings with a similar typology. The building information has been extracted from the original project which has been consulted in several municipal archives. Then, a specific analysis was carried out, in which the seismic weaknesses of the building have been identified. An analysis of the operation and the behaviour of the building has also been carried out.

After this first analysis, different reinforcement models using various seismic retrofitting solutions (steel cross bracings and steel jackets) have been calculated and compared. This research is focused on obtaining and comparing the capacity curves and the performance points of different retrofitting solutions. In addition, it is important to note that the study has been performed in a real building, in contrast with other studies that use only theoretical models. Because of this, the result of this study can be extrapolated to other buildings with a similar

typology and a real intervention project could be made in future works. Finally, it must be highlighted that, compared with previous researches this article try to obtain the most effective retrofitting scheme considering structural, architectural and constructive parameters.

The school building and the different seismic retrofitting techniques have been analysed by calculating the limit state of damage according to the EC-08 [10]. In this analysis, the performance point has been used to set the level of damage expected. The Significant Damage (SD) limit state, in which the structure is significantly damaged and it is not economically viable to repair it, has been used for a comparison.

The rest of this paper is structured as follows. First, the state of art is described: the most relevant studies on the seismic vulnerability of buildings are mentioned. Afterwards, the methodology used to determine the seismic vulnerability and to evaluate the different seismic retrofitting solutions is outlined. Then, the case study building and its seismic weak points are described in detail. Next, the results of the different seismic retrofitting solutions analysis are presented. Lastly, the conclusions for this study are summarised.

## State of the art

Apart from school buildings, the seismic vulnerability of many other typologies has also been analysed. In those studies, the systematic evaluation of a large number of buildings in different cities or countries has been done. For example, several seismic vulnerability and risk research works have been performed on an urban scale in big cities such as Barcelona [11, 12] or Lisbon [13]. The seismic behaviour of buildings in Barcelona (Spain) [11, 12] was analysed via the capacity spectrum. In that case, the seismic vulnerability, the risk, the number of casualties and the economic losses were evaluated. [13] studied the seismic vulnerability and the risk analysis of old buildings of a neighbourhood in Lisbon. Three popular Portuguese typologies were analysed using a pushover analysis, obtaining the capacity curves. After that, the fragility curves of the models were calculated, estimating the building damage and the economic and life losses for different earthquake scenarios.

Regarding the studies centred on school buildings, many schools are located in high seismicity areas. Due to this, several countries such as Japan [14], Venezuela [15] and Italy [16] have developed ambitious programmes in order to increase the seismic resilience of schools. In the case of Venezuela [15], school buildings were analysed and retrofitted. In Venezuela there are many school buildings in highly hazardous regions, and it is necessary to carry out a seismic vulnerability analysis. For instance, four school buildings collapsed during the 1997 Cariaco earthquake. In [15], a national programme to evaluate and reduce the seismic risk in existing schools was described. [17] analysed the public-school buildings in Istanbul using a probabilistic structural fragility. This analysis is compared with damage ratios calculated for similar building typologies for the Istanbul building inventory. Other researchers have analysed the building damage that has been caused by a real earthquake. For example, the Lorca earthquake in 2011 [18] and the Japan 2011 earthquake [19].

In this research, different seismic codes which present different seismic analyses and retrofitting techniques have been used. The ATC-40 [20] presents different seismic retrofitting solutions based on increasing the building strength, reducing the earthquake demand or increasing the deformation capacity. The FEMA 356 [21] points out the following rehabilitation techniques: local modification of components, removal or lessening of existing irregularities, global structural stiffening, global structural strengthening, mass reduction, seismic isolation and supplemental energy dissipation. Finally, the EC-08 part 1 [10] and part 3 [22] have been used to consult the seismic analysis method and the seismic retrofitting procedure. Several seismic retrofitting solutions are presented in these documents.

Many RC frame buildings constructed in southern European countries have been designed before modern seismic codes existed. These buildings are potentially vulnerable to seismic lateral forces. Therefore, there is a lot of research on seismic rehabilitation techniques which set out different seismic retrofit systems. [23] evaluated the effectiveness of three different retrofitting solutions (RC jacketing, steel bracing and concrete shear wall). To do so, numerical models with non-linear static analyses were used. Furthermore, the seismic performance was compared with the non-linear dynamic analyses results. As a conclusion, the steel bracings reduced the displacement demand, and increased the global deformation and dissipation capacities of RC buildings. However, the connections between the brace members and the structure produced high stress concentrations. In addition, the steel bracings inserted symmetrically and along the perimeter of the buildings can increase the torsional stiffness of torsional-unbalanced structures.

There is a large number of studies about seismic retrofitting techniques applied to theoretical models and laboratory analysis [24]. However, there are few studies focused on the application of these techniques on real buildings. In addition, in this research, architectural integration and the ease and speed of construction are considered. The seismic retrofitting of school buildings has been reported in a few studies. In [25], an optimisation technique has been implemented to obtain the minimum number and the location of seismic retrofitting in concrete columns required for school buildings. This work concluded that a staggered positioning of the retrofitted columns is more effective than continuous positioning. In addition, the location of retrofitting in different areas to each floor contributes to economic benefits.

Steel bracings are one of the most efficient solutions for seismic retrofitting in RC frame buildings. [26] analysed several seismic retrofitting techniques with different configurations (X braced, inverted V braced, ZX braced, and Zipper braced) through a static non-linear push-over analysis. Three-storey and six-storey buildings with different steel bracing configurations were studied. The research concluded that brace retrofitting enhances the global capacity of the buildings in terms of strength, deformation and ductility. Moreover, the X and Zipper bracing systems performed better according to the type and the size of the cross section.

There are many studies on seismic retrofitting with steel jackets [27, 28] and steel braces [29–33]. Several of these research works analyse the brittle failure connection between the brace and the building, which is this system's main weakness. Others have analysed the materiality of the braces because conventional steel braces have a buckling failure in some cases. For instance, in [34] the non-compression carbon fibre X-bracing system was analysed. They concluded that this system increases the strength and it has no buckling failure, but this system must be improved, especially in connection with the original structure. In this research, conventional steel X-bracings have been used because of their low cost and easy construction.

In [35], the following factors have been analysed: (i) the behaviour of RC columns before and after retrofitting with steel X-bracing, (ii) the possible complications, the increase in demand and the side effects of the latter retrofitting method. The study concludes that in high-rise frame buildings (in high-rise buildings, for frames located near the top), this retrofitting system has adverse effects on those columns that are attached to the bracing elements. As a general conclusion, retrofitting low-rise RC frames with steel X bracing is beneficial to the performance of columns in almost every aspect.

Several studies applied seismic retrofitting to a real model. In the paper [36], a number of seismic retrofitting systems (steel eccentric braces, steel buckling retrained braces and steel shear panel) were tested statically and dynamically. The analysis was performed in a real RC building located in Bagnoli (Naples, Italy). The results showed the effectiveness of the metal systems analysed in order to improve the strength, stiffness and ductility of the RC structure's original capacity.

Among the majority of research related to steel bracing, this retrofitting system shows important positive effects, which increase shear capacity, reduce displacements and decrease drifts.

The analysis of seismic vulnerability in urban spaces is a very critical line of research for society because it helps to create resilient communities. There are many studies concerning the seismic behaviour of a large number of buildings, i.e., in cities, neighbourhoods, etc. Among the most relevant to the present work are the aforementioned studies in the cities of Barcelona and Lisbon. Although most of the research has been carried out in residential buildings, there are also studies that specifically target school buildings, in particular the cases of Venezuela and Istanbul, cities which were mentioned earlier and are worth noting. For all the reasons discussed earlier, school buildings are the most vulnerable and, therefore, seismic analysis in school buildings should be carried out more accurately and more extensively.

In these research studies, the most common method for the analysis of the seismic vulnerability of buildings is the capacity-spectrum method and nonlinear static analysis. Pushover analysis has been selected since it is the most appropriate considering the main goal of the research and the building school scale [37]. A dynamic analysis would be more accurate but would also require more time, greater computational effort and data than the static procedures.

The majority of these studies mainly performed experimental analyses. The different seismic retrofitting systems were analysed in theoretical or laboratory models. There is a lack of studies which analyse the seismic retrofit behaviour in real buildings. There are several studies about the effectiveness of the X-bracing or steel jacket retrofitting systems in theoretical models or residential buildings, but these retrofitting systems are not applied to school buildings.

In the present work, two different retrofitting techniques (steel jacket and steel X-bracing) have been used. Several retrofitting schemes have been assessed and compared by means of the pushover analysis of a real case study building. The analysis of results in terms of capacity and probability of damage enabled obtaining important conclusions about the impact of these retrofitting schemes in the building's seismic behaviour.

## Methodology

In this section, the different methodology steps proposed in this paper are presented. First, the different steps in the seismic vulnerability analysis proposed (building information, structural model, analysis pushover, etc.) are explained. Then, the constructive and structural characteristics and the case study building´s configuration are presented. Next, the seismic weak points present in the structure are set out. Finally, the results of the seismic retrofitting techniques are discussed.

### Methodology overview

Firstly, the data of the primary school building (blueprints and characteristics) are obtained from Local Archives. The structural system (slab type, storey height, span length, etc.), constructive characteristics and project configuration are retrieved from the original project. A database has been created with the building information needed in the analysis (general dimensions, structural and constructive characteristics, etc.). The general geometry of the structure is then reflected in a 3D stick model, including the existing building elements (column, beam, joint, waffle slabs, etc.).

Secondly, the 3D model is imported into SAP2000 v.20 commercial software [38], where the different structural elements (columns, beams, slabs and joint) are accurately defined, including element sections and material models. The waffle slabs have been modelled as finite

element shells which have been defined with the bidirectional slab characteristics. The frames' elements (beams and columns) are modelled as a linear element which has elastic and nonlinear properties. These elements are defined with a section which has dimensions, materials and steel rebar characteristics. The nonlinear behaviour of RC elements has been simulated by adding plastic hinges as in [39]. The hinges have been defined according to ASCE-41-13 [40] failure criteria. Regarding their location, the hinges have been added at the ends of both beams and columns (5% and 95% of the total element length), as recommended in the EC-8. M3 type plastic hinges have been selected for beams in order to consider bending. In the case of columns, PM2M3 type plastic hinges have been selected to take into account the axial force and biaxial moment.

The selection of an appropriate load distribution is an important factor in static pushover analysis [37]. According to EC-8 [22], two load patterns have been considered in this analysis. The first is proportional to the mass and height of the slab at each storey and it is introduced as a linear distribution with an inverted triangle. The second is proportional to the displacement of each storey in the predominant elastic mode of vibration. For this type of buildings, both load patterns yield fairly similar results [37]. The control node needs to be situated in the centre of the mass in the top floor of the building.

The infills are included in the model due to their potential (negative or positive) effect on the seismic performance of the building [41], both in terms of seismic demand and capacity, especially given their layout's lack of symmetry. Moreover, the presence of partial infills in height in the main façade of some of the buildings studied could cause short-column effects (Fig 4C). The infills have been put into the model in accordance with [42]: with two crossed diagonal braces which work only in compression, induced by lateral loads.

In this study, the seismic performance of the building is defined by its Performance Point (PP), which is the point in the shear-force / displacement domain, where the building capacity meets the seismic demand. The PP is obtained by the capacity-demand spectrum method [43]. In order to perform this method, both building capacity curves and response spectra are needed. The PP, which represents the maximum response of the structure, has been obtained by intersecting the response spectrum and the capacity curves, both in spectral coordinates, according to the N2 methods [44], using the iterative procedure proposed in annex B of the EC-8 part 1 [10].

The capacity curves graph the relationship between the base shear force and the displacement of a building's roof. In this research, a nonlinear pushover analysis has been used in order to obtain these curves, since it is more appropriate for the project's scale and the paper's main goal. In addition, although dynamic analyses are more accurate, they require greater computational effort, data and time than static procedures [45]. The capacity curves have been obtained first for the as-built configuration, and later for the retrofitted building, with different retrofitting schemes. Finally, the capacity curves have been analysed and compared graphically.

The response spectrum is defined according to EC-8 and the Spanish annex [46]. This is determined by several parameters: the $a_{gr}$ which is the peak ground acceleration on type A ground, the type of soil and the importance of the building. In that sense, the basic acceleration for Spain has been selected from the Updated Seismic Hazard Map of Spain 2012 [47]. Then, in the EC-8 there are five types of soil, whose values affect the response spectrum. The type of soil is obtained through geotechnical studies, depending on the place where the building is located.

In the next phase, the seismic safety assessment of both the original structure and the different retrofitted structural schemes is performed. The damage has been evaluated by means of the PP and considering the different limit states according to the EC-8 part 3 [22]. The

ratio "performance point" / "limit state displacement" of the structure has been analysed and compared with the results of the different seismic retrofit systems. According to the EC-8 part 3 [22] there are three limit states: Limit State of Near Collapse (NC), Limit State of Significant Damage (SD), and Limit State of Damage Limitation (DL), and also considering the limit state for operationality (OP), which is presented in the Italian Code NTC 2018 [48] and will probably be in the future generation of the Eurocodes. The damages suffered by the building have been identified with the damage limit states. These points are identified on the capacity curve of the school building as a fuction of the yielding ($d_y$) and the ultimate displacement ($d_u$). In this case, the OP, DL, SD and NC are equal to $S_{d1}$, $S_{d2}$, $S_{d3}$ and $S_{d4}$ respectively. These damage limit states have been calculated and we have applied the following functions (1), which correspond to slight ($S_{d1}$), moderate ($S_{d2}$), severe ($S_{d3}$) and complete ($S_{d4}$) seismic damage, respectively, the same equations used by [12]. These equations have been implemented to calculate the damage limit state from the idealised bilinear capacity curve of the building.

$$S_{d1} = 0,7d_y$$
$$S_{d2} = d_y$$
$$S_{d3} = d_y + 0,25(d_y + d_u)$$
$$S_{d4} = d_u$$

(1)

All of the information related to the building (capacity curves, building's location, structural characteristics, etc.) are imported in the specific software, which has been developed in the PERSISTAH project to evaluate the seismic vulnerability of building´s schools [49]. The aim is to obtain the different points of damage limit states ($S_{d1}$ (OP), $S_{d2}$ (DL), $S_{d3}$ (SD), $S_{d4}$ (NC)) (Fig 2), the PP displacement ($d_t$), fragility curves and the probability percentage of the damage limit state. The fragility curves have been obtained through the HAZUS software methodology [50], which is the method implemented in the specific software. The fragility

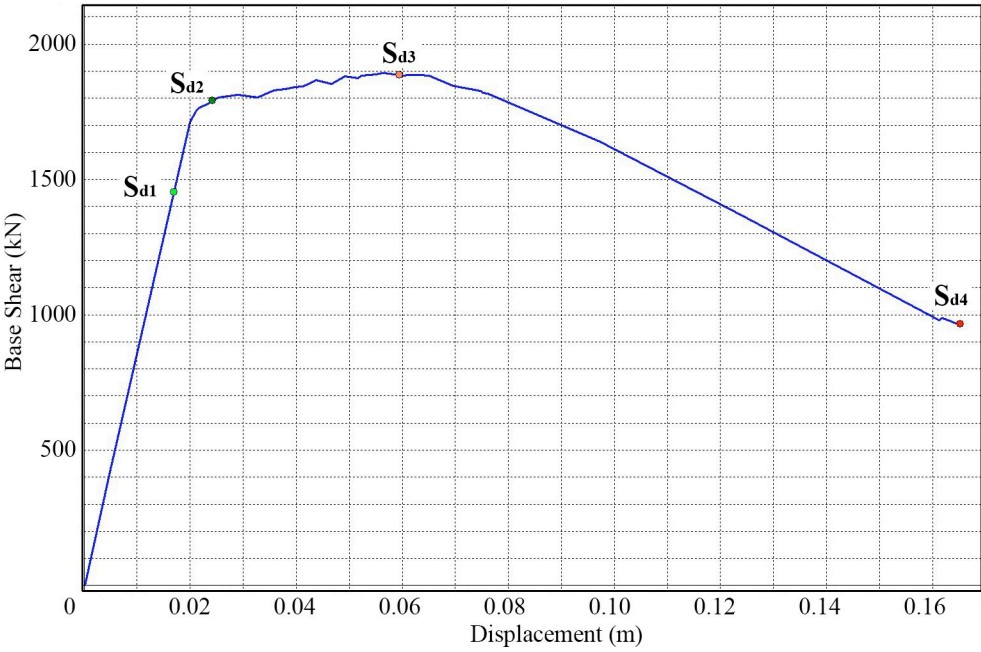

**Fig 2. Specific software.** Capacity curve with three limit state displacements according to EC-08.

curves are computed for each PP. These curves are defined by a lognormal probability distribution and are constructed according to the spectral displacement. The structural fragility is the probability of the damage level exceeding a give limit state, for a given ground motion level. The probability percentages of different limit states have been obtained by means of the PP spectral displacement and the fragility curves.

The seismic safety is evaluated with the SD ($S_{d3}$) limit state. The structure at this level presents significant damage while still capable of some residual lateral strength and stiffness, i.e. it can sustain after-shocks of moderate intensity. The repair cost of the structure at this point is uneconomic. The PP displacement ($d_t$) is compared with the displacement ($d_{SD}$) corresponding to the SD limit state. If ($d_t$) is lower than ($d_{SD}$), then the structure is safe. If ($d_t$) is higher than ($d_{SD}$), then the structure is unsafe. The condition ($d_t < d_{SD}$) is checked in all seismic retrofit systems to confirm that the structure is safe in the event of an earthquake. Finally, the retrofitting resistance of each model is compared with its corresponding capacity curves and PPs. As a result, the improved seismic behaviour is analysed and compared.

The results obtained calculating the structure with a multiple-degree-of-freedom (MDOF) model are converted to their equivalent single-degree-of-freedom (SDOF) system through the mass equivalent to SDOF ($m^*$) (2) and the transformation factor ($\Gamma$) (3), according to annex B of the EC-8 part 1 [10].

$$m^* = \Sigma m_i \phi_i = \Sigma F_i \tag{2}$$

$$\Gamma = \frac{m^*}{\Sigma m_i \phi_i^2} = \frac{\Sigma F_i}{\Sigma \left( \frac{F_i^2}{m_i} \right)} \tag{3}$$

The limit states of damage are defined considering the idealised elasto-perfectly plastic force-displacement relationship, according to annex B of the EC-8 part 1 [10]. This is obtained by implementing the pushover analysis, which has been explained earlier, to obtain the building's capacity curve. The capacity of the equivalent non-linear SDOF system is modelled as a bilinear capacity curve (Fig 3). The yield strength ($F_y^*$), which represents the ultimate strength

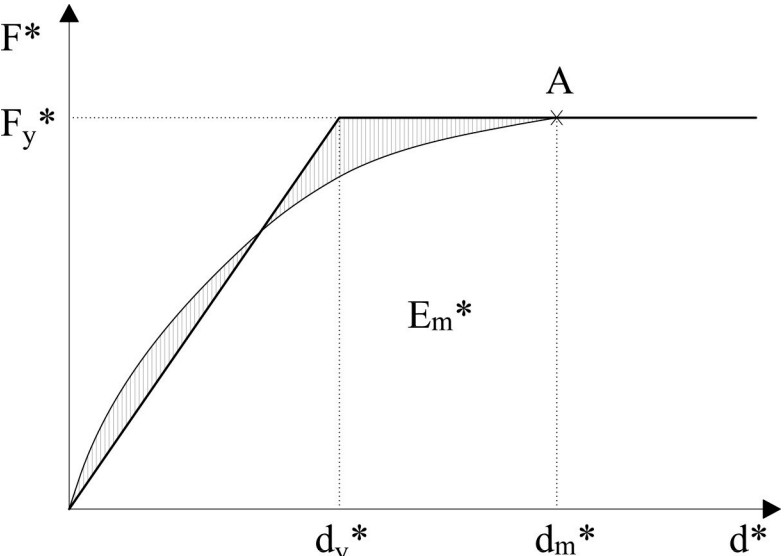

**Fig 3. Bilinear capacity curve.** SDOF system equivalent.

of the idealised system, is equal to the base shear force at the formation of the plastic mechanism. In this sense, the yield displacement of the idealised SDOF system ($d_y^*$) is given by the equation (9). Where $d_m^*$ is the displacement and $E_m^*$ is the deformation energy leading to the formation of the plastic mechanism (A).

$$d_y^* = 2\left(d_m^* - \frac{E_m^*}{F_y^*}\right) \tag{4}$$

The period of the structure of the equivalent SDOF system is calculated according to the following function (10) which is in accordance with annex B of the EC-8 part 1 [10].

$$T^* = 2\pi\sqrt{\frac{m^* d_y^*}{F_y^*}} \tag{5}$$

Once this first analysis of the building's seismic behaviour has been carried out, the most critical points where the structure shows a weak seismic behaviour are identified. In this case, the formation of mechanisms in the plastic hinges of some zones of the structure are analysed, as well as the structure displacement.

The seismic weak points detection is an important step with the aim of having a first approach to the possible failure points of the structure in case of earthquake. This will help to choose the seismic retrofitting typology and the priority areas in the structure to insert it. In the 2011 Lorca earthquake [18], the most damaged buildings had severe failures in these seismic weak points and several non-structural elements. For this reason, a first specific visual analysis to identify weak element typologies of the structure (short columns, short beams, waffle slabs, soft storeys, etc.) is performed. Then, those element typologies that present a greater weakness according to the previous analysis are selected for retrofitting. For each of them, the seismic retrofit scheme has been selected according to the seismic behaviour and seismic weak points analysis. These retrofitting schemes are included in the model to improve their seismic behaviour. Finally, the resulting schemes for each retrofit solution are analysed and compared.

At this point, it is assessed that the structure aided by all the different seismic retrofit models is safe in terms of its damage limit states. The different retrofitting systems are analysed according to their seismic effectiveness, their degree of architectural integration in the building and their level of construction simplicity. This methodology can be used in other future studies. In Fig 4, the process followed in this method has been illustrated graphically.

## Case study building

**Building description.**   In the current study, a school building, which is one of the most common typology in the Huelva region, has been analysed in the present work. The building is located in Huelva, whose soil type is III according to the Spanish seismic construction code of building (NCSE-02) and is type C according to the EC-8. Furthermore, the schools are class importance III according to EC-8 part 1 [10]. In this case, the basic acceleration has been multiplied with the importance factor $\gamma_I = 1.3$, according to the Spanish annex to the EC-8 [46].

In particular, there are 13 schools that share the exact same blueprints (Table 1). Due to this, the seismic retrofit proposal could be extrapolated to a wide range of schools with a similar structural system. Furthermore, it should be taken into account that the 13 schools' locations have a different Peak Ground Acceleration (PGA), according to the updated Seismic Hazard Map of Spain 2012 [47]. For this study, a school located in Huelva city is selected, as it has the highest PGA: 0.12.

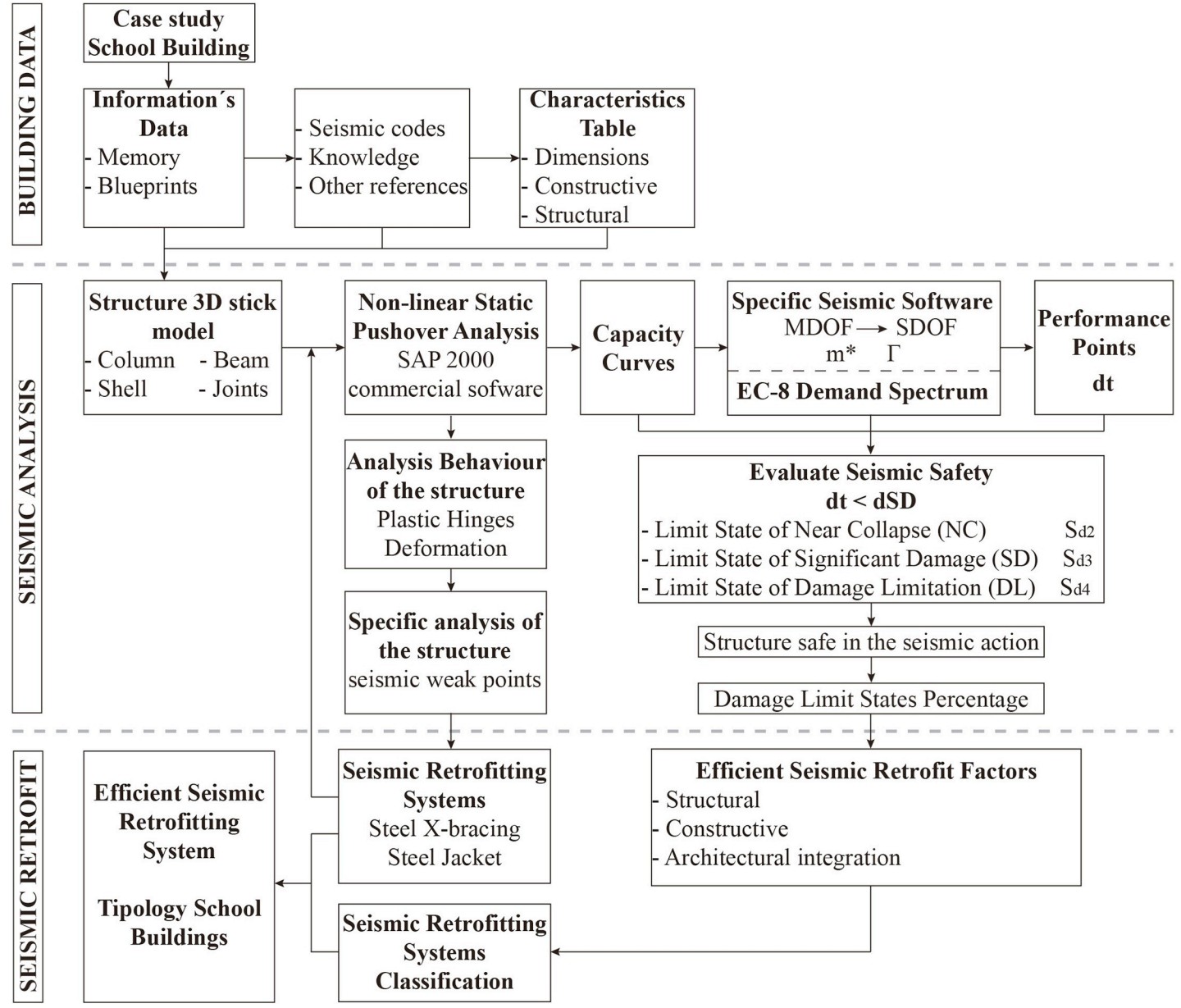

**Fig 4. Schematic diagram of the methodology used to perform the proposed seismic vulnerability and retrofit analysis.**

The case study building is a three-storey RC frame building that has an H-shape floor plan (Figs 5 and 6). This building was constructed in 1976. The common areas are located on the ground floor with a central hall and an exterior porch (Fig 5A). The ground floor suspended slab is a ribbed slab and the other levels are waffle slabs. The classrooms are located in the outer part of the bays, whereas the corridors, stairs and toilets are situated in the inner part (Fig 5B).

The ground floor suspended slab is 0.65 m high, whereas the other floors are 3.3 m high. This is based on a double symmetry design with three parallel strips and a central yard which is covered with a canopy. The floor, which has a dimension of 45.32 x 21.60 m, is configured through a module of 0.9 x 0.9 m. This has a central structural joint which divides it into two

**Table 1. School building with the same compact typology in Huelva region (Spain).**

| School | Location | PGA (TR = 475) |
| --- | --- | --- |
| CEIP Los Llanos | Almonte | 0.10 |
| CEIP José Romero Macías | Aroche | 0.07 |
| CEIP Las Viñas | Bollullos Par del Condado | 0.10 |
| CEIP Manuel Pérez | Bollullos Par del Condado | 0.10 |
| CEIP Divino Salvador | Cortegana | 0.07 |
| CEIP Fuenteplata | Gibraleón | 0.10 |
| CEIP José Oliva | Huelva | 0.12 |
| CEIP Marismas del Odiel | Huelva | 0.12 |
| CEIP Onuba | Huelva | 0.12 |
| CEIP Oria Castañeda | Lepe | 0.12 |
| CEIP Maestro Rojas | Nerva | 0.07 |
| CEIP José Nogales | Valverde del Camino | 0.08 |
| CEIP Los Rosales | Huelva | 0.12 |

structural blocks of 22.5 x 21.6 m. It has ten spans parallel to the X-direction and three spans parallel to the Y-direction. In the X and Y directions, the span between columns is 4.5 m and 7.5 m respectively (Fig 7A). In this research only one of the structural blocks (Fig 7B) has been calculated and analysed since both blocks are exactly symmetrical.

The RC frames characteristics are shown in Table 2. The columns have a rectangular cross-section with a dimension of 0.30 x 0.45 m in all the storeys. All the beams are 0.30 m wide and 0.30 m deep and the thickness of the waffle slab is 0.30 m. The ribbed floor slab has a thickness of 0.22 m. The concrete is H-200 and the steel rebar AEH-400. This data has been consulted in the original project of the school.

The gravitational loads (GL) have been obtained from the school´s data and the Spanish code CTE [8], the weight of structural and constructive elements (W) is: bidirectional slabs, thickness $<0.30$m ($4 kN/m^2$), the ceramic flooring $<0.08$m ($1 kN/m^2$), the roof of tiles gables over lightened partitions ($3 kN/m^2$), the internal partitions ($1 kN/m^2$), the ceramic flooring ($1 kN/m^2$), the infills ($10 kN/m$) and the live load (Q) for public spaces with tables and chairs ($3 KN/m^2$). These have been combined according to the combination GL = W+DL+0.3Q established in the seismic code NCSE-02 [51].

**Seismic weak points.** Firstly, a visual analysis of the building has been carried out. It is important to acquire a thorough understanding of the floor plan and façade configuration (Fig 5A and 5C). The building has several weak points which are characterised by a poor aseismic behaviour. The seismic retrofit system should integrate smoothly in the building and it ought not to affect its use. Also, the façade openings should not be affected negatively by the retrofit system.

The building is composed of waffle slabs and flat beams, which nowadays is a very common structural system. This typology has a low ductility value according to the NCSE-02 Spanish code [51]. Several seismic codes do not recommend these structural elements in seismic areas. In [52], an analysis regarding buildings with low ductility is carried out. In this research, the behaviour and the overload stress values in a three-storey building are calculated and evaluated. The waffle slabs show a high probability of severe damage compared to a building designed in accordance with the current Spanish codes EHE-08 [53] and NCSE-02 [51].

The building has, on the one hand, a ground floor suspended slab and, on the other hand, horizontal openings in the façade which do not cover the entire column length. Because of this, there are short columns which have a weak aseismic behaviour. These structural elements

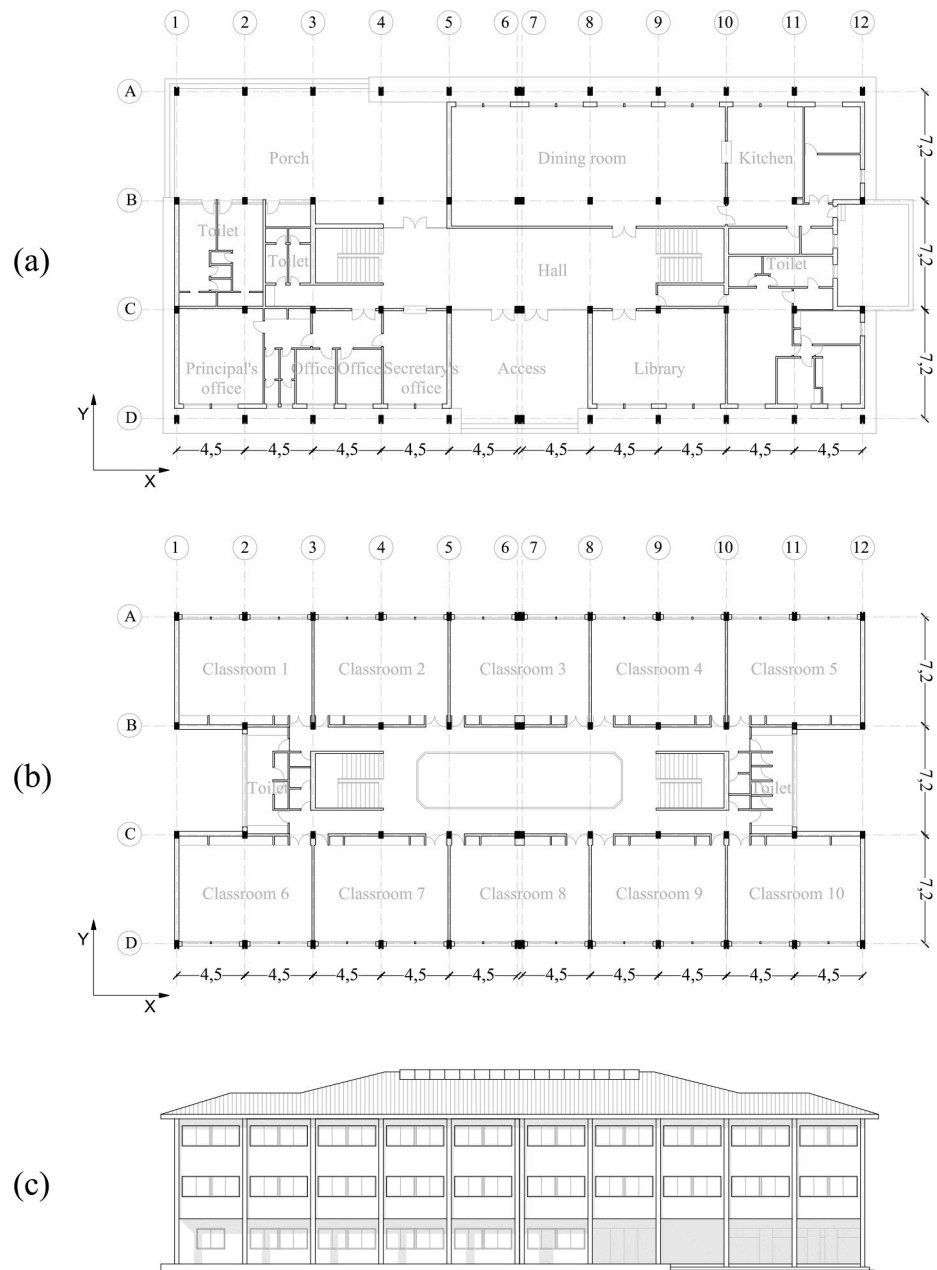

**Fig 5.** School´s distribution on the ground floor (a), first-second floors (b) and façade (c).

receive high shear stress and they may result in double diagonal cracking failure when they receive lateral loads during an earthquake. Additionally, the ground floor has an elevated percentage of isolated columns (Fig 5A). Due to this, the ground floor is less stiff than the upper floors and, in the case of an earthquake, will suffer more deformation, with early formation of plastic hinges at the top and bottom of the columns. Furthermore, the floor plan distribution of these isolated columns is rather irregular, which may cause a strongly asymmetric seismic behaviour. Due to the presence of the central structural joint, the lateral displacement of the building during an earthquake may mean that the two structural blocks collide with each other. This would cause damage in the infill walls and structural elements. In the 2011 Lorca

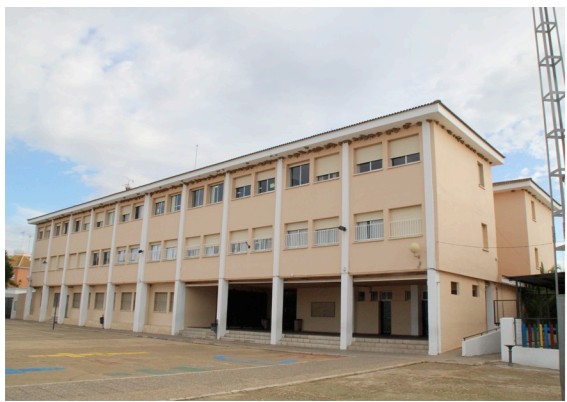
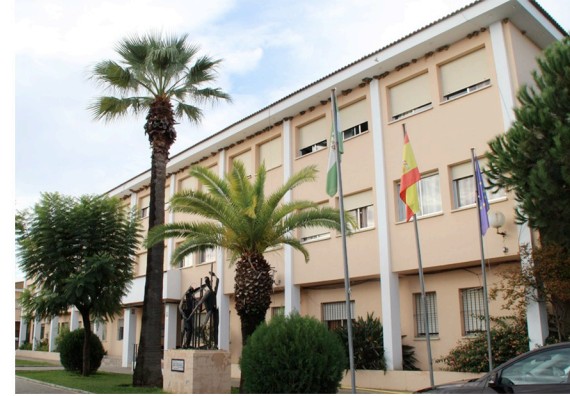

**Fig 6. Pictures of the school´s typology selected as case study (author's ownership).**

Earthquake [18], many failures were derived from both weak structural elements (short columns and soft floors) and the collapse of different non-structural elements.

Based on the analysis of the weak points of the building, two seismic retrofitting techniques have been selected to study their effectiveness, (i) Steel X-Bracing, and (ii) Steel Jackets (Fig 8). The selection of these retrofitting techniques has been selected considering the following aspects: constructability, architectural impact (both aesthetics and functional), cost and hindrance to educational activity.

These retrofitting techniques have been configured in several retrofitting models, as shown in Figs 9–11. The purpose of the steel bracing solution is to counter the deformation and displacement in ground floor columns. This system requires strengthening and stiffening according to the ATC-40 [20] and has been implemented mostly in the X-direction where stresses

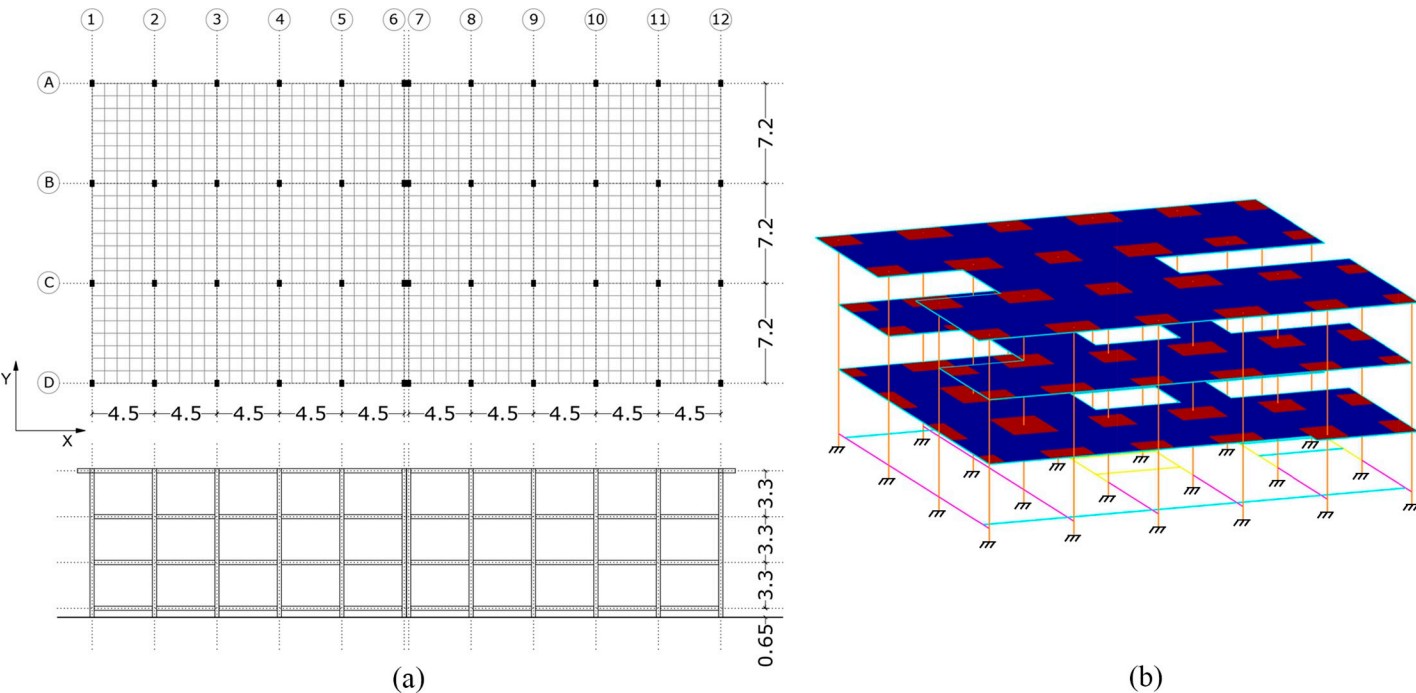

**Fig 7. School´s configuration (dimensions in metres).** (a) Dimensions and module configuration (b) 3D model.

**Table 2. Parameters of structural elements.**

| Parameters | Columns | Load Beams | Tied Beams | Tied Beams Bidirectional Slabs |
|---|---|---|---|---|
| | | Unidirectional Slab | Unidirectional Slab | |
| Dimensions | 30x45 cm | 30x50 cm | 30x30 cm | 30x30 cm |
| Longitudinal rebar | 6Ø16mm | Top: 4Ø16mm | Top: 2Ø16mm | Top: 2Ø16mm |
| | | Lower: 4Ø16mm | Lower: 2Ø16mm | Lower: 2Ø16mm |
| Transversal rebar | Ø6/22cm | Ø6/25cm | Ø6/15cm | Ø6/15cm |

showed higher values and where a higher number of plastic hinges formed. Steel bracing is one of the most popular systems of seismic retrofitting in RC frames. The construction of this method adds less mass to the structure and it can be constructed with less disruption of the building. In addition, the construction of this method in the building is easy because the columns on the ground floor are isolated and they are accessible for the assembly of the steel braces.

Steel Jackets have been selected due to the presence of short columns and a system that enhances deformation capacity according to the ATC-40 [20] has been applied. They are inserted in the first and second floors to improve the seismic behaviour of short columns. This seismic retrofit improves the shear capacity and seismic deformation behaviour in these structural elements. In addition, in the façade, part of the columns' profile is not bounded by infill elements (Fig 8). Due to this, this part is accessible for the construction of the steel jacket in the column. This system covers the exterior area of the structural element with a steel jacket, providing an easy construction and good architectural integration (Fig 8).

## Retrofitting models definition

For the seismic retrofitting proposals of the case study building, and according to the weak points and the seismic behaviour of the original building of the model, two different options were analysed under several scenarios: X-Bracing (XB) and Steel-Jacket (SJ) reinforcement

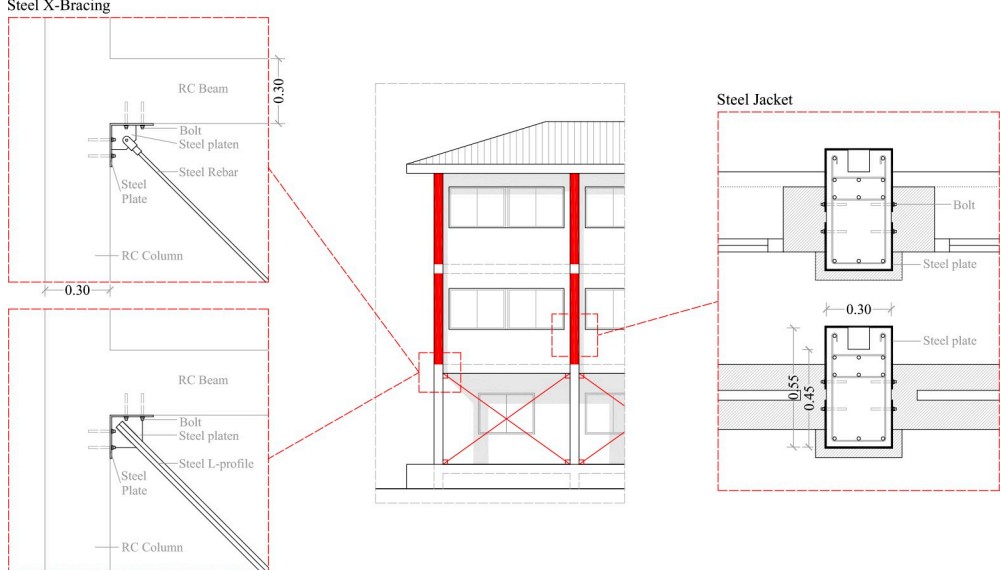

**Fig 8. Constructive detail of the retrofitting solutions.**

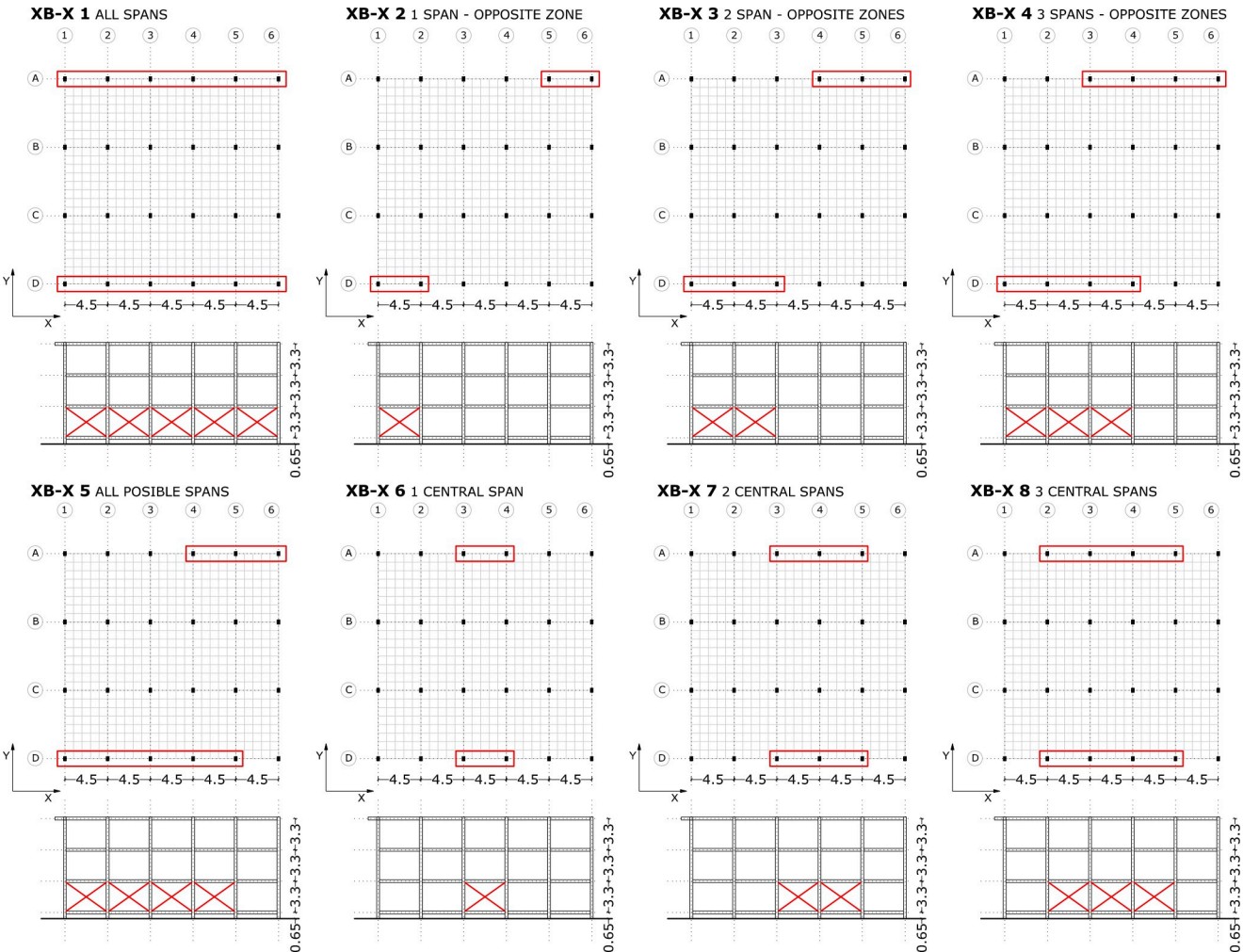

**Fig 9. Seismic retrofitting models with X-bracing.** X-direction.

(Fig 7). These retrofitting techniques have been considered with the aim of improving the building's seismic behaviour and the seismic weak points detected. They have been analysed in different schemes in the school building (Figs 6, 8 and 9), where the retrofitting elements have been situated with different configurations, varying the position and the number of them.

The Steel L-profile (50*50*3mm) and steel rebar of ø25 mm section have been used in the X-bracing elements. The steel jacket has been incorporated with a 3 or 5 mm thick steel plate. The rebar material is steel B400S. Its elastic limit ($F_y$) is 400 MPa and the modulus of elasticity ($E_c$) is 200,000 MPa. The L-profile material is structural steel S275. Its unit weight is 76.98 kN/m3, the modulus of elasticity ($E_c$) is 210,000 MPa and the elastic limit ($F_y$) is 275 MPa.

**X-Bracing retrofitting scenarios definition.** The name of the different retrofitting models has been determined according to the different procedure. The retrofitting element is assigned: X-Bracing (XB) or Steel Jacket (SJ). Then, in case of XB the following value is the X or Y direction. The end factor is the number of the retrofitting model. In the following figures (Figs 9 and 10), XB retrofitting schemes have been represented.

**Steel-jacket retrofitting scenarios definition.** The different SJ retrofitting models have been named with a number of retrofitting scenarios. In the following figure (Fig 11), SJ retrofitting schemes have been represented.

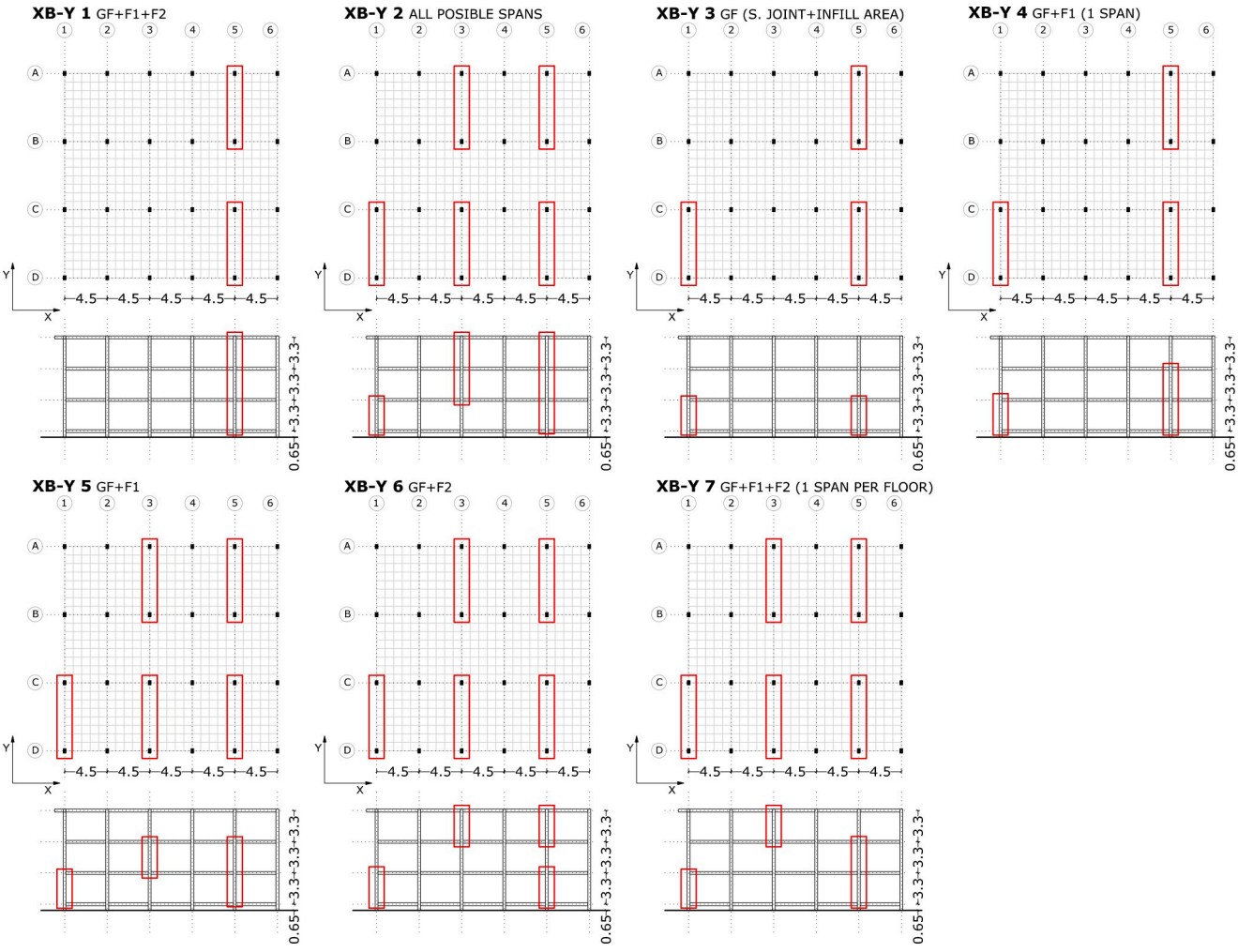

**Fig 10. Seismic retrofitting models with X-bracing.** Y-direction.

## Architectural integration of seismic retrofitting models

The architectural integration of the seismic retrofitting regards not only the visual impact of the intervention, but also, and quite importantly, that the seismic retrofit systems do not interfere with the building's configuration and use. In this light, the X-bracing integration is better than RC shear walls because isolated columns populate the ground floor (Fig 5A). This seismic retrofitting does not interfere with the openings of the façade. In addition, this system does not reduce visibility and illumination in the interior building. Another important factor is the ease of construction and the reduction of the execution time, because it is very important that the teaching period is not interrupted. In the X-direction, the installation of X-bracing systems is quick and easy to perform as they can be fixed to isolated columns (Fig 8).

In the Y-direction, the seismic retrofit with X-bracing has been introduced in the façade and interior wall spans without openings. In this case, the X-bracings are integrated in the interior of walls so that the different openings do not interfere.

The seismic retrofitting with a steel jacket has been incorporated into columns in the first and second floor. In this case, the construction is more complicated but this seismic retrofitting is integrated completely in the building. It is introduced by covering the columns with a metal sheet (Fig 8).

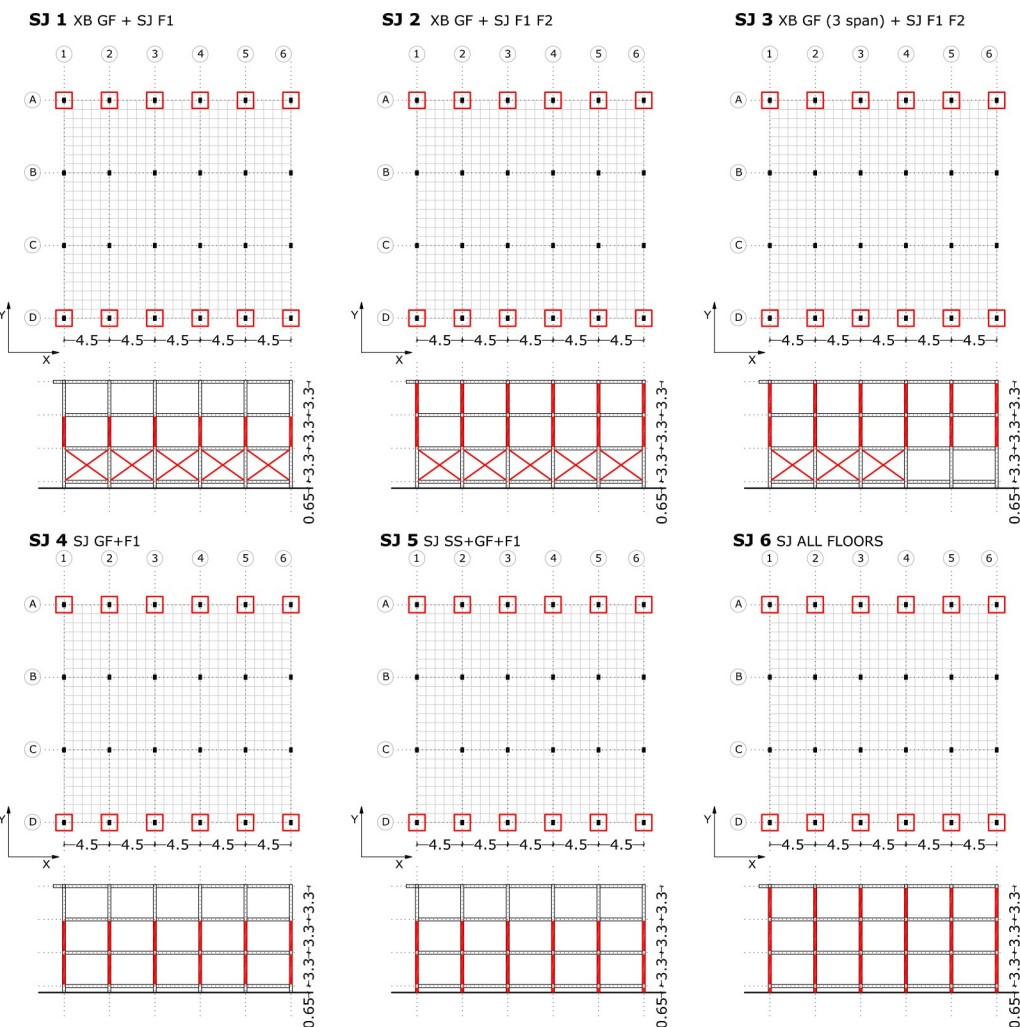

**Fig 11. Seismic retrofitting models with steel jackets.**

## Results and discussion

The most relevant results obtained from all the hypotheses calculated are shown in this section. First, the calculated main periods of vibration are discussed. Second, an analysis of the structure deformation and plastic hinges formation in structural elements is presented. Third, the structure's capacity is analysed by means of the capacity curves and the PPs of different retrofitting models. Then, an analysis of structural security against the seismic action by means of damage limit states is laid out. Finally, an analysis of probability of damage limit states is presented.

### Period of vibration

The main period of vibration of the original building was found to be T = 1.022s in X direction. After the retrofitting (SJ3), this period shifted to T = 0.906s. Regarding the Y direction, the original period was of T = 0.84s, decreasing down to T = 0.73s after retrofitting (XB-Y 2).

### Structure deformation analysis

In the first analysis, the building's deformation and formation of plastic hinges in some structure zones have been analysed in the structural elements. The default plastic hinge properties

are implemented according to FEMA 356 [21]. The ultimate rotation capacity of structural elements is defined by different deformation states: Immediate Occupancy (IO), Life Safety (LS) and Collapse Prevention (CP). They have a plastic hinge deformation capacity of 10%, 60% and 90%, respectively.

In XB-X 1 (Fig 9), all the spans have been reinforced by steel X-bracing (L-profile and steel ø25 section), in the X-direction. This model improves the column deformation and the number of plastic hinges is reduced in the ground floor. Only plastic hinges in the range IO-LS appear in the ends of columns. Nevertheless, the deformation of first floor columns increases. The majority of plastic hinges are concentrated in columns of this floor. Furthermore, the short columns, which have plastic hinges in the residual strength (C-D) range fail by shear force.

In XB-X 4 and XB-X 8 (Fig 9), the results are similar. In these cases, the majority of the plastic hinges, which are in the C-D range, are concentrated in columns in the first floor. The short columns also fail by shear force.

In XB-X 3 and XB-X 7 (Fig 9), with a steel L-profile, the majority of the plastic hinges are concentrated in the ground floor columns in the range C-D (residual strength). In this case, the two central spans provide a greater improvement in the formation of plastic hinges than the opposite spans. In the case of X-bracing (ø25 section), the reduction of deformation in ground floor columns is better than using an L-profile in the previous two models. The façade columns have plastic hinges in the IO-LS range and the interior columns have plastic hinges in the C-D range. In this case, the first floor columns have a high number of plastic hinges and the short columns fail by shear force.

In general, the different seismic retrofit models with steel X-bracing improve the ground floor column deformation and upgrade the structure's seismic behaviour. However, the first floor columns have a larger deformation and they present several plastic hinges in both ends. The short columns in this floor have a high deformation and develop plastic hinges in both ends in the range of residual strength. Due to this, it is necessary to introduce steel jackets in these columns to enhance the seismic behaviour, solving the short column problem.

In the second retrofit model, steel jackets with different thicknesses have been introduced in the upper floor columns (Fig 11). In SJ 1, the number of plastic hinges in the first floor columns is reduced, in contrast to the second floor short columns which have plastic hinges in the C-D range.

The analysis shows that in general, the different retrofit models using only steel jackets (SJ4, SJ5 and SJ6) have similar results. The ground floor and ground floor suspended slab column deformation is similar to the original RC structure. The majority of plastic hinges, which are in the C-D range, are concentrated in the ends of ground floor columns. In the SJ 4 (Fig 11) with different thickness jackets, several short columns in the ground floor suspended slab have collapsed.

Retrofitting models with steel X-bracing in the Y-direction (Fig 10) improve the performance of the structure in general, and in particular, its irregular deformation (rotation). The XB-Y 3 model (Fig 10) benefits the reduction in deformation of ground floor columns. In contrast, the first floor columns present greater deformation and more plastic hinges in the C-D range (residual strength) at the top and bottom.

In the XB-Y 1 model (Fig 10), the columns in the façade show a greater deformation than those in the reinforced spans area. The majority of plastic hinges are concentrated in the ends of ground floor columns. Furthermore, several columns in the façade have developed plastic hinges at the top, in the range of C-D. The deformation in the façade columns, which have restricted movement by infill elements, is (0.016–0.114) (0.098m).

The XB-Y 4 and XB-Y 5 models (Fig 10) reduce the formation of plastic hinges in the columns of all the floors. However, several ground floor columns have a plastic hinge residual strength range (C-D) at the top. In the case of XB-Y 6 (Fig 10), the majority of plastic hinges are at the top and bottom in the first floor columns, which have a greater deformation (0.0343 to 0.1039m) (0.0696m). The XB-Y 7 scheme (Fig 10) considerably reduces the formation of plastic hinges in the different floor columns.

## Structure capacity analysis

There is a significant difference in the building's capacity curves for each direction. The building presents a higher capacity in the Y-direction, with a base shear of 3325 KN for displacements (0.05285m). In the X-direction, the base shear is 1877 KN for higher displacement (0.06615m). This could be due to fact that the Y-direction is the direction of the higher inertia of the columns and the loading direction of the ground floor suspended slab. The structure is symmetric and, because of that, the retrofit zones have been defined with symmetric configurations so that the seismic structure deformation is regular.

In the X-direction, the columns in the ground floor suffer a greater deformation (13 cm) than the in the upper floors, which have a deformation of (0.2 cm). Due to this, the greatest concentration of plastics hinges is at the top and bottom columns of the ground floor in the range of residual strength (range C-D). Furthermore, several short columns in the ground floor suspended slab have collapsed. This problem has been resolved with steel X-bracing retrofitting which improves the structure's seismic behaviour, increasing the strength and stiffness of frames and preventing the formation of soft-stories.

In the X-direction, the structure is unsafe against the seismic action (Table 3). The PP displacement ($d_t = 0.064$m) is greater than the Limit State of Significant Damage ($d_{SD}$). In this direction it is necessary to introduce seismic reinforcement so that the structure is safe. The structure has a high percentage in the limit of damage $S_d3$ and $S_d4$ (Fig 12A).

$$d_t = 0.064\text{m} \quad d_{SD} = 0.059\text{m} \quad d_t > d_{SD}$$

In the Y-direction, some columns in the ground floor present a very large deformation (6.65 cm). The concentration of plastics hinges is at the top and bottom columns of the ground floor in the range of residual strength (range C-D). The structure has rotated, and due to this, the columns near the structural joint present a higher deformation. This behaviour can be explained by the fact that in this area there are no bracing infills, which restrict movement in these columns.

**Table 3. Damage limit states displacement and performance points (X-direction).** X-bracing (L-steel profile 50*50*3mm).

| Seismic retrofit model | Damage Limit States displacement (m) | | | | | Performance Point | |
| --- | --- | --- | --- | --- | --- | --- | --- |
| | Sd1 | Sd2 | Sd3 | Sd4 | | dt (m) | Ft (KN) |
| Original | 0.017 | 0.025 | 0.059 | 0.162 | **UNSAFE** | 0.066 | 1877.78 |
| XB-X 1 | 0.035 | 0.051 | 0.062 | 0.094 | **SAFE** | 0.051 | 3518.56 |
| XB-X 8 | 0.038 | 0.054 | 0.071 | 0.123 | **SAFE** | 0.059 | 3265.70 |
| XB-X 7 | 0.050 | 0.071 | 0.090 | 0.146 | **SAFE** | 0.067 | 2993.35 |
| XB-X 6 | 0.030 | 0.042 | 0.073 | 0.163 | **SAFE** | 0.068 | 2449.97 |
| XB-X 4 | 0.038 | 0.054 | 0.070 | 0.119 | **SAFE** | 0.060 | 3295.21 |
| XB-X 3 | 0.060 | 0.086 | 0.107 | 0.169 | **SAFE** | 0.066 | 2981.00 |
| XB-X 2 | 0.028 | 0.040 | 0.070 | 0.159 | **SAFE** | 0.069 | 2452.82 |
| XB-X 5 | 0.048 | 0.068 | 0.092 | 0.162 | **SAFE** | 0.063 | 3008.08 |

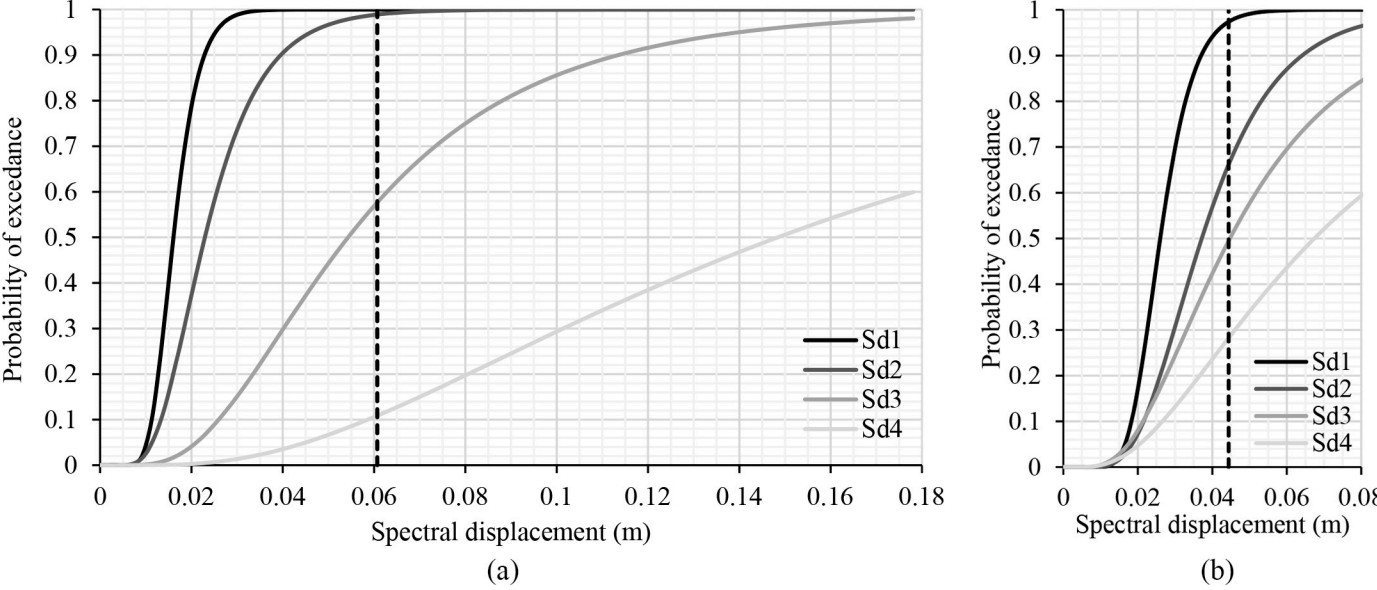

**Fig 12. Fragility curves.** (a) X-direction (b) Y-direction.

In the Y-direction, the structure is safe in the event of an earthquake (Table 6). The PP displacement ($d_t$) is smaller than the Limit State of Significant Damage ($d_{SD}$). However, the PP is very close to the limit state $d_{SD}$. The structure reaches a high probability percentage in the limit of damage $S_d3$ and $S_d4$ (Fig 12B).

$$d_t = 0.053m \quad d_{SD} = 0.055m \quad d_t < d_{SD}$$

Thus, several seismic retrofitting proposals have been studied in order to lower the PP and the probability percentage in the $S_d3$ and $S_d4$ limit of damage.

The results of the different retrofitting models analysed show the effectiveness of the steel-based reinforcement systems examined in improving the original capacity of the RC structure in terms of strength and stiffness. Figs 13A, 14A, 15A and 16A illustrate the comparison among capacity curves and Figs 13B, 14B, 15B and 16B show a comparison of the PPs corresponding to the different seismic retrofitting models tested, also showing an improvement in the response with respect to the original RC structure.

Fig 13A compares the capacity curves of the different seismic retrofitting with X-bracing (L-steel profile 50*50*3mm) in the X-direction. In general, the increase in the number of reinforced spans increases the building's capacity. As expected, the strongest capacity has been obtained with this XB-X 1, XB-X 4 and XB-X 8 retrofitting system, which produces a large increase of stiffness and strength.

The capacity and PPs (Fig 14A and 14B) are similar in those models that share the same number of retrofit spans, in opposite or central spans. The XB-X 5 model has a similar shear force to those other models where only two spans are reinforced (XB-X 3 and XB-X 7), whereas the PP and displacement is slightly lower. One possible explanation for this is that the model has a different number of retrofit spans in each façade, thus the model has a worse aseismic behaviour.

In the case of models with X-bracing (steel ø25 section) in the X-direction, Fig 14A illustrates that a bigger capacity is obtained with smaller displacements than in the case of X-bracing with (L-steel profile 50*50*3mm) (Fig 13A). When reinforcing all spans, the PP in the first

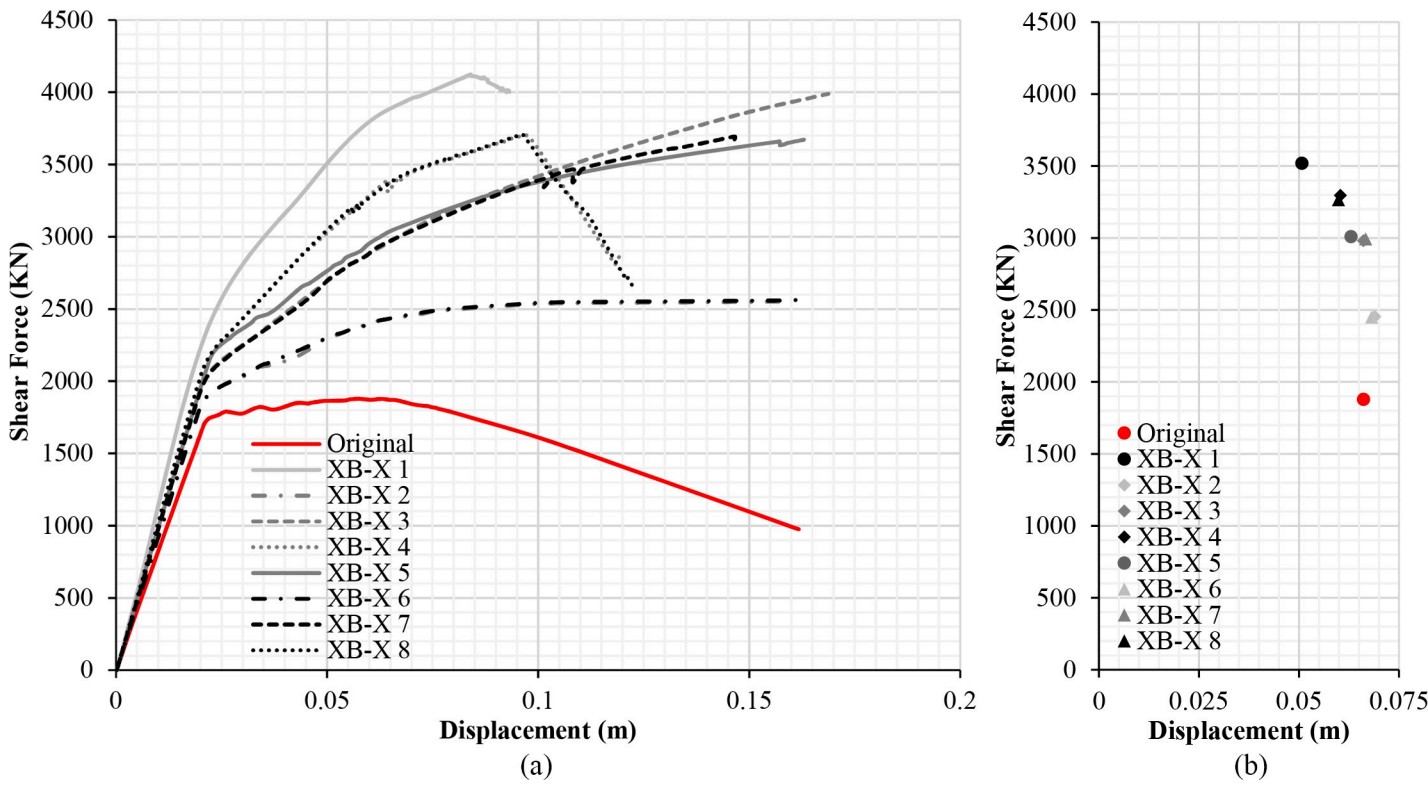

**Fig 13.** Capacity curves (a) and performance points (b) (X-direction). X-Bracing (L-steel profile 50*50*3mm).

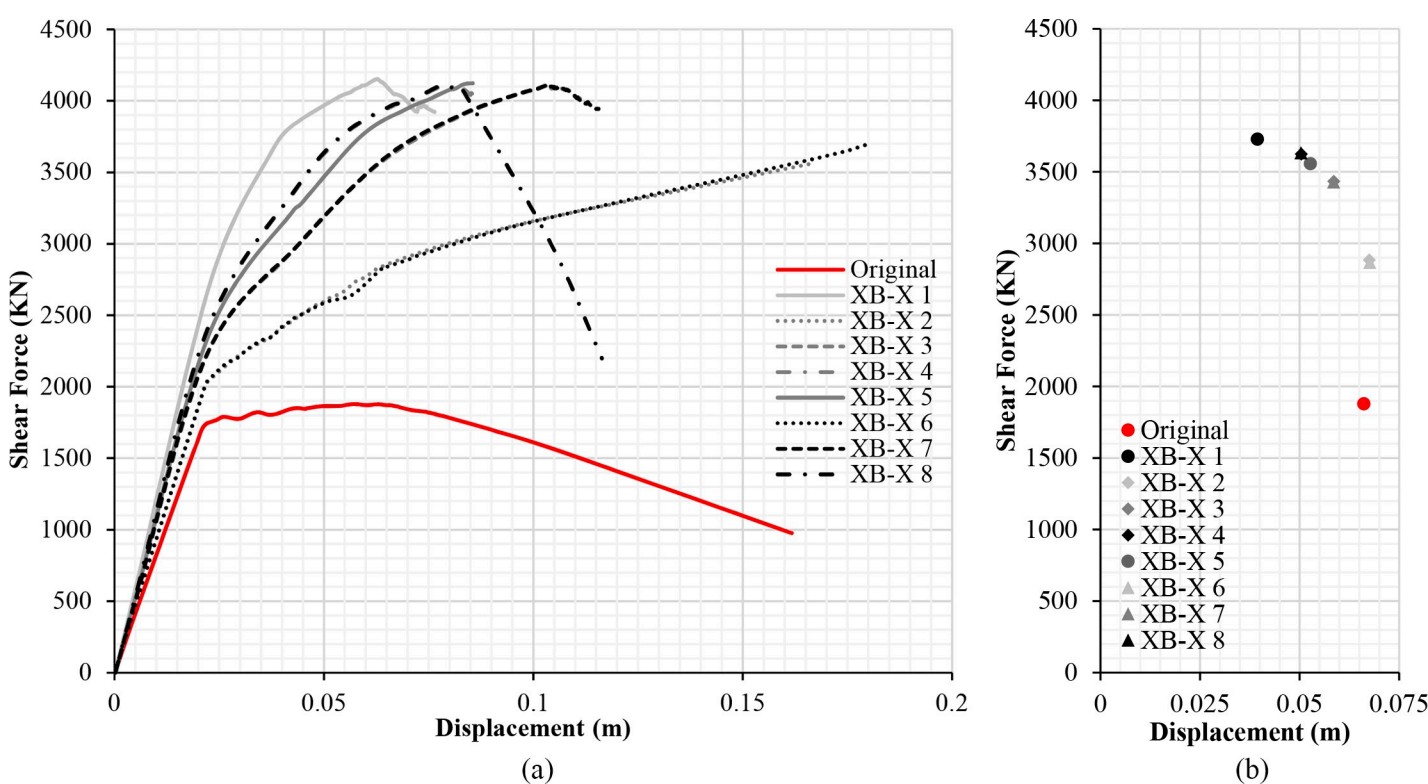

**Fig 14.** Capacity curves (a) and performance points (b) (X-direction). X-bracing (steel ø25 section).

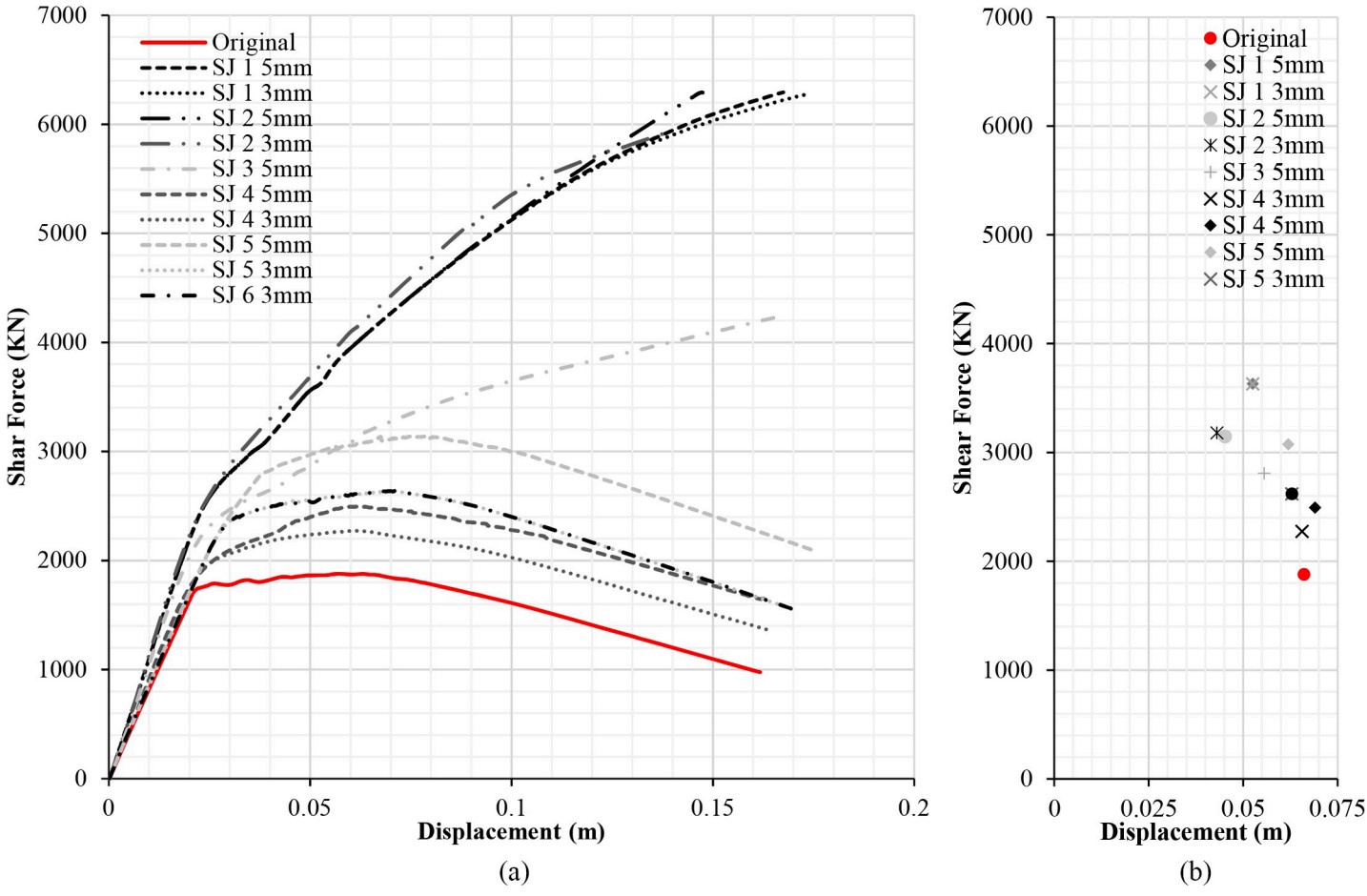

**Fig 15.** Capacity curves (a) and performance points (b) (X-direction). X-Bracing + Steel Jacket.

case is $d_t = 0.03943$ (Fig 14B), which is fairly lower than the displacement in the second case $d_t = 0.05073$ (Fig 13B).

Fig 15A displays the capacity curves of different retrofitting models with X-bracing and steel jackets in the X-direction. As expected, the biggest capacity has been obtained in retrofit systems with steel X-bracing on the ground floor (SJ 1, SJ 2 and SJ 3), which produce a large increase of capacity. Little variation in the capacity has been observed in the retrofitting models which only use steel jackets, even after testing several thicknesses and configurations (SJ 4, SJ 5 and SJ 3). The capacity of the SJ 5 5mm model is the highest of all the retrofitting models with steel jackets. It is relevant to emphasise that both the model where all the first floor columns are reinforced and the one where all the columns in the first and second floors are reinforced present the same capacity (SJ 1 and SJ 2). However, reinforcement should be introduced in any case in the first and second floors because both are affected by short column problems which need to be addressed.

Fig 16A depicts the capacity curves of retrofit models with X-bracing (steel ø25 section) in the Y-direction. In this direction the building presents a higher capacity than in the opposite direction since it is the direction of higher inertia of the columns. In general, all the retrofit models improve the capacity of the structure. The XB-Y 5 model has a higher capacity than the XB-Y 6 model given the same displacement (Fig 16A), despite having the same number of reinforced spans. However, the XB-Y 6 scheme has a greater irregularity in the distribution of

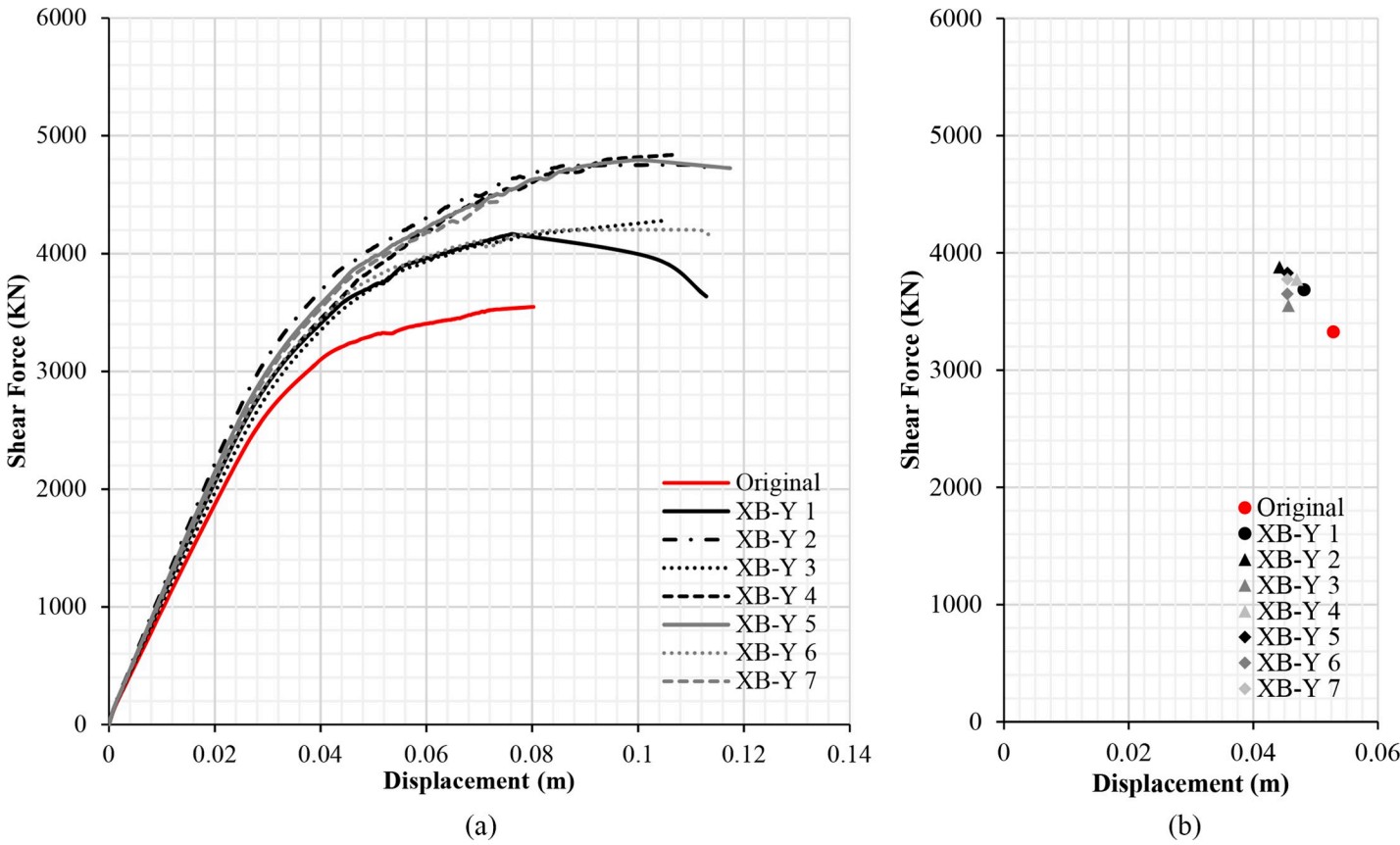

**Fig 16.** Capacity curves (a) and performance points (b) (Y-direction). X-bracing (steel ø25 section).

the retrofitting spans than XB-Y 5. The XB-Y 2 model has the greatest capacity and PP (Fig 16B). Several models (XB-Y 5, XB-Y 6 and XB-Y 7) present the same displacement with varying shear force at the PP. In general, all the retrofitting models improve the PP which increases the shear force with smaller displacements (Fig 16B).

## Damage limit states displacement and structure security

Tables 3–6 show the displacements of the different damage limit states and the PPs of the original RC structure and the seismic retrofitting models. The results in Tables 3 and 4 show that in

**Table 4. Damage limit states displacement and performance points (X-direction).** X-bracing (steel ø25 section).

| Seismic retrofit model | Damage Limit States displacement (m) | | | | | Performance Point | |
|---|---|---|---|---|---|---|---|
| | Sd1 | Sd2 | Sd3 | Sd4 | | dt (m) | Ft (KN) |
| Original | 0.017 | 0.025 | 0.059 | 0.162 | UNSAFE | 0.066 | 1877.78 |
| XB-X 1 | 0.028 | 0.040 | 0.049 | 0.076 | SAFE | 0.039 | 3727.89 |
| XB-X 8 | 0.034 | 0.048 | 0.065 | 0.116 | SAFE | 0.050 | 3632.93 |
| XB-X 7 | 0.042 | 0.060 | 0.074 | 0.115 | SAFE | 0.058 | 3429.21 |
| XB-X 6 | 0.060 | 0.085 | 0.109 | 0.179 | SAFE | 0.068 | 2866.49 |
| XB-X 4 | 0.034 | 0.049 | 0.058 | 0.087 | SAFE | 0.050 | 3623.41 |
| XB-X 3 | 0.042 | 0.060 | 0.073 | 0.112 | SAFE | 0.059 | 3433.18 |
| XB-X 2 | 0.052 | 0.075 | 0.098 | 0.165 | SAFE | 0.067 | 2883.67 |
| XB-X 5 | 0.036 | 0.052 | 0.060 | 0.085 | SAFE | 0.052 | 3557.05 |

**Table 5. Damage limit states displacement and performance points (X-direction).** X-bracing + Steel Jacket.

| Seismic retrofit model | Sd1 | Sd2 | Sd3 | Sd4 | | dt (m) | Ft (KN) |
|---|---|---|---|---|---|---|---|
| | **Damage Limit States displacement (m)** | | | | | **Performance Point** | |
| Original | 0.017 | 0.025 | 0.059 | 0.162 | **UNSAFE** | 0.066 | 1877.77 |
| SJ 1 3mm | 0.074 | 0.106 | 0.123 | 0.173 | **SAFE** | 0.052 | 3630.53 |
| SJ 1 5mm | 0.074 | 0.106 | 0.121 | 0.167 | **SAFE** | 0.052 | 3630.53 |
| SJ 2 3mm | 0.062 | 0.089 | 0.102 | 0.140 | **SAFE** | 0.052 | 3614.17 |
| SJ 2 5mm | 0.072 | 0.104 | 0.114 | 0.147 | **SAFE** | 0.050 | 3699,32 |
| SJ 3 5mm | 0.060 | 0.086 | 0.106 | 0.166 | **SAFE** | 0.052 | 3614.17 |
| SJ 4 3mm | 0.021 | 0.030 | 0.063 | 0.164 | **UNSAFE** | 0.065 | 2274.09 |
| SJ 4 5mm | 0.025 | 0.035 | 0.067 | 0.163 | **UNSAFE** | 0.069 | 2492.48 |
| SJ 5 3mm | 0.024 | 0.034 | 0.068 | 0.169 | **SAFE** | 0.062 | 2618.56 |
| SJ 5 5mm | 0.029 | 0.042 | 0.075 | 0.175 | **SAFE** | 0.062 | 3074.00 |
| SJ 6 3mm | 0.024 | 0.034 | 0.068 | 0.169 | **SAFE** | 0.062 | 2618.56 |

general the seismic retrofit models with steel X-bracing improve the seismic behaviour of the structure in the X-direction. In this case, the structure is safe in all the retrofitting models. The displacements are shorter and the base shear is greater in those models with a steel ø25 section (Table 4) than X-bracing with a steel L-profile (Table 3).

In the case of the seismic retrofitting models with X-bracing + Steel Jackets (Table 5), the models with steel X-bracing in the ground floor produce the largest increase in base shear with the smallest top displacement. There is little improvement in the PP in those retrofitting models which only use steel jackets. The PP displacements (dt) with these reinforcement models is near to or surpasses the limit state displacement ($S_{d3}$). Specifically, the structure is unsafe in SJ 4 of 3 and 5 mm models.

The seismic retrofit models with steel X-bracing in the Y-direction (Table 6) improve the seismic behaviour of the structure in all the schemes. In this case the structure is safe in all the retrofitting models.

## Probability percentage of different damage limit states

Figs 17–19 show the comparison of probability percentage of the damage limit state of the different seismic retrofitting models. Also, they show the improvement of probability with respect to the original RC structure.

In the case of X-bracing (steel L-profile 50*50*3mm and ø25 section) in the X-Direction, Fig 17A and 17B, all the models improve the probability of damage of the original RC

**Table 6. Damage limit states displacement and performance points (Y-direction).** X-bracing (steel ø25 section).

| Seismic retrofit model | Sd1 | Sd2 | Sd3 | Sd4 | | dt (m) | Ft (KN) |
|---|---|---|---|---|---|---|---|
| | **Damage Limit States displacement (m)** | | | | | **Performance Point** | |
| Original | 0.031 | 0.044 | 0.053 | 0.080 | **SAFE** | 0.053 | 3325.10 |
| XB-Y 1 | 0.034 | 0.048 | 0.064 | 0.113 | **SAFE** | 0.048 | 3684.43 |
| XB-Y 2 | 0.037 | 0.054 | 0.068 | 0.113 | **SAFE** | 0.044 | 3878.17 |
| XB-Y 3 | 0.038 | 0.054 | 0.067 | 0.105 | **SAFE** | 0.045 | 3550.20 |
| XB-Y 4 | 0.043 | 0.061 | 0.073 | 0.107 | **SAFE** | 0.047 | 3772.23 |
| XB-Y 5 | 0.040 | 0.058 | 0.073 | 0.117 | **SAFE** | 0.045 | 3827.55 |
| XB-Y 6 | 0.034 | 0.049 | 0.065 | 0.113 | **SAFE** | 0.045 | 3649.26 |
| XB-Y 7 | 0.035 | 0.051 | 0.057 | 0.074 | **SAFE** | 0.045 | 3774.33 |

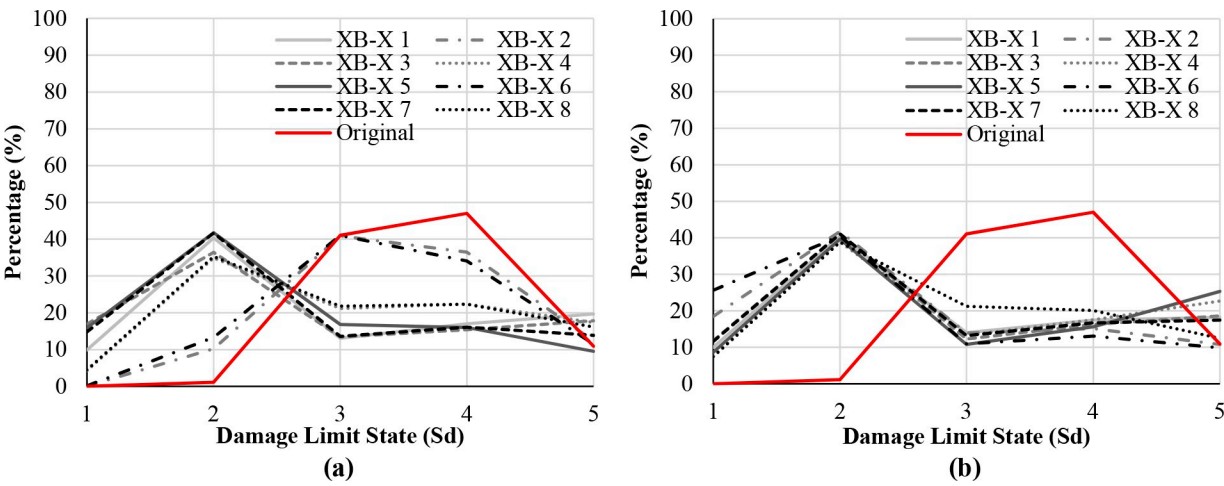

**Fig 17. Damage limit states percentage X-bracing in the X-direction.** L-steel profile 50*50*3mm (a) and steel ø25 section (b).

structure. In general, the higher percentages have been obtained in the damage limit state Sd2 (Damage Limitation (DL)) which is approximately 40%. In this damage limit state, the structure is only lightly damaged, according to EC-08 part 3 [22]. The structural elements prevent significant yielding and retain their strength and stiffness properties. The XB-X 2 solution with a steel L-profile (50*50*3mm) does not cause significant changes in the probability of damage regarding the original RC structure. The larger percentage is in Sd3 (40%) and the percentage in Sd4 is also high (35% approximately).

In the case of X-bracing + steel jacket in X-Direction (Fig 18), those seismic retrofitting models with only steel jackets do not improve the probability of damage significantly with respect to the original RC structure. In these models, the percentage in $S_{d4}$ has been reduced and the percentage in $S_{d2}$ and $S_{d3}$ has been increased slightly. The percentages of the damage limit states $S_{d3}$ and $S_{d4}$ are very high in general. In the models with X-bracing in the ground

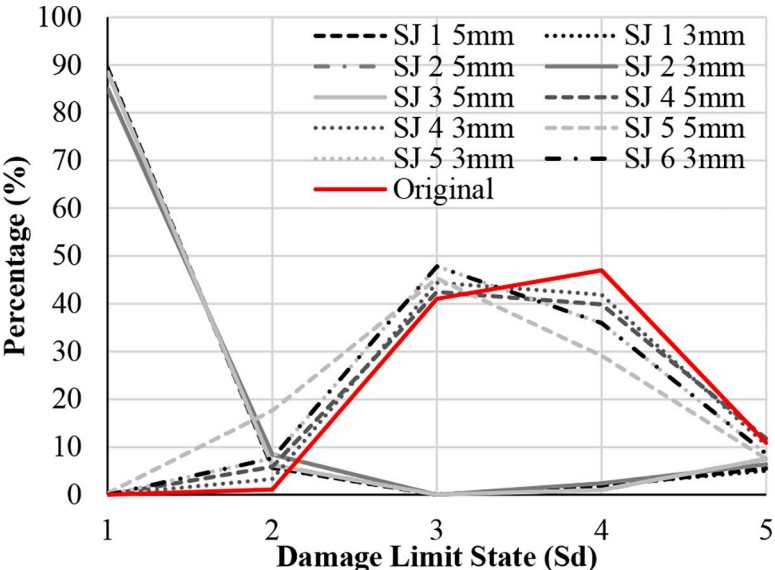

**Fig 18. Damage limit states percentage in the X-direction.** X-bracing + Steel Jacket.

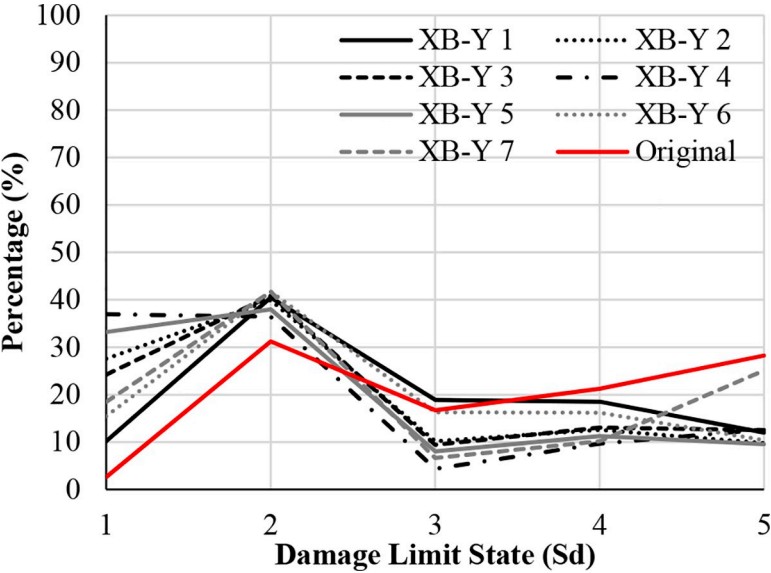

**Fig 19. Damage limit states percentage in the Y-direction.** X-bracing (steel ø25 section).

floor and steel jackets in the upper floor, the probability of damage in the structure has been improved significantly. The highest percentage is in Sd1 (85%).

In the case of X-bracing (steel ø25 section) in the Y-Direction (Fig 19), all the models increase the probability percentage in the damage limit states $S_{d1}$ and $S_{d2}$. The highest percentages have been obtained in $S_{d2}$ (Damage Limitation (DL)), which is approximately 40%. Therefore, the probability percentage of the damage limit states $S_{d3}$ and $S_{d4}$ has been reduced by approximately 5–10%.

## Conclusions

This RC compact school typology is one of the most common typologies of school buildings in Huelva. This typology is vulnerable, and it is unsafe in a seismic event. Therefore, a seismic retrofitting solution is necessary to reduce its vulnerability. A retrofit system with steel X-bracing and steel jackets has been selected, upon thorough analysis, from a range of models. This seismic retrofitting solution improves the seismic behaviour and reinforces the seismic weak points detected in this structure.

The analysis of the results shows that the seismic reinforcement of the structure is necessary to guarantee the security of the school in the case of an earthquake. It has also been demonstrated that without retrofitting, the structure is unsafe under seismic loads in the X-direction. In the Y-direction, the PP displacement is near the damage limit state $S_{d3}$, therefore the structure can suffer much damage during an earthquake. In this case, the analysis has demonstrated that the structure could be safe provided that the seismic retrofitting proposed is applied.

The results have shown that the steel X-bracing models produce the greatest improvements in the structure's seismic behaviour. The X-bracing solution in the X-direction improves the seismic behaviour of the ground floor columns, which present a greater deformation in the original RC structure. This retrofitting intervention increases the strength and stiffness of the frames and prevents the formation of soft-storey effects in the ground floor. The X-bracing model with a steel ø25 section conveys a greater reduction in displacements in comparison with the steel L-profile 50*50*3mm model.

The analysis has shown that steel jacketing in the upper floors in the X-direction is necessary to solve the short column problem, which is a seismic weak point. Furthermore, the deformation in the first floor columns increases when steel X-bracing is introduced in the ground floor. In this case, the majority of the plastic hinges are concentrated in the first floor columns in the residual strength range.

In general, the capacity and the seismic damage probability are considerably improved in all the seismic retrofit models. The highest percentages have been obtained in $S_{d2}$ which correspond to DL (Damage Limitation).

The most efficient retrofitting model is XB-X 2 and XB-X 3 because it enhances the structure's seismic behaviour and it does not interfere with the functioning of school buildings. Furthermore, the probability that the building is in the damage limit state of Damage Limitation (DL) $S_{d2}$ increases up to the maximum percentage (38%, approximately). The retrofitting distribution (opposite or central zones) does not affect the increase in capacity, since different retrofit models share very similar results. The seismic capacity of the structure increases according to the number of reinforced spans in each façade.

The capacity of the structure increases slightly in the retrofitting models with steel jackets of different thicknesses. Several of them present high percentages in the damage limit state $S_{d3}$ in which the structure is unsafe against seismic actions. However, the retrofitting schemes with X-bracing in the ground floor and steel jacket in the upper floor produce a remarkable improvement and the high percentages are in $S_{d1}$ (90%, approximately).

The seismic retrofitting with steel jackets in the second floor columns does not improve the capacity of the structure. However, the seismic retrofit is introduced in all columns of both floors to solve the problem of short columns. Although the upper floors are reinforced with steel jackets, the ground floor must also be reinforced with steel X-bracing so that the structure improves its seismic behaviour and capacity.

As a result, it has been demonstrated that selecting the weak seismic points where adding the seismic retrofitting elements is more effective than obtaining a profitable improvement in the structure. In that sense, the most effective solutions have been proved to be the retrofitting scheme SJ3 with two or three reinforced spans in the ground floor by means of X-bracing in the X-direction, which is the most vulnerable in the building. These retrofitting proposals do not interfere in the use of the building and they considerably enhance the seismic performance of the original structure.

In future studies, the ease of construction of this seismic retrofitting model will be analysed. A quick and easy construction technique will be developed. The construction will be subsequently extrapolated to 13 school buildings with the same typology in the zone.

## Supporting information

**S1 File.**
(DWG)

## Author Contributions

**Conceptualization:** Antonio Morales-Esteban.

**Data curation:** Emilio Romero-Sánchez.

**Formal analysis:** Emilio Romero-Sánchez, Antonio Morales-Esteban.

**Funding acquisition:** Antonio Morales-Esteban.

**Investigation:** Emilio Romero-Sánchez, Antonio Morales-Esteban, María-Victoria Requena-García-Cruz.

**Methodology:** Emilio Romero-Sánchez, Antonio Morales-Esteban, María-Victoria Requena-García-Cruz.

**Project administration:** Antonio Morales-Esteban.

**Resources:** Emilio Romero-Sánchez, Antonio Morales-Esteban.

**Supervision:** Antonio Morales-Esteban, Beatriz Zapico-Blanco, Jaime de-Miguel-Rodríguez.

**Validation:** Emilio Romero-Sánchez, Antonio Morales-Esteban.

**Visualization:** Emilio Romero-Sánchez.

**Writing – original draft:** Emilio Romero-Sánchez, Antonio Morales-Esteban.

**Writing – review & editing:** Antonio Morales-Esteban, Beatriz Zapico-Blanco, Jaime de-Miguel-Rodríguez.

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
