## [Decision Letter · Decision Letter 0]

10 Jul 2020

PONE-D-20-15119

Specific seismic retrofitting of a compact reinforced concrete building with X-bracings and steel jackets. Application to a primary school in Huelva.

PLOS ONE

Dear Dr. Morales-Esteban,

Thank you for submitting your manuscript to PLOS ONE. After careful consideration, we feel that it has merit but does not fully meet PLOS ONE’s publication criteria as it currently stands. Therefore, we invite you to submit a revised version of the manuscript that addresses the points raised during the review process.

We look forward to receiving your revised manuscript.

Kind regards,

Francisco Martínez-Álvarez

Academic Editor

PLOS ONE

Journal Requirements:

3. Please clarify in your Data availability statement where the data from the study can be found. Please list both the original sources as well as information on how other researchers can access the data.

4. We note that Figures 4, 5, 6 in your submission contain copyrighted images. All PLOS content is published under the Creative Commons Attribution License (CC BY 4.0), which means that the manuscript, images, and Supporting Information files will be freely available online, and any third party is permitted to access, download, copy, distribute, and use these materials in any way, even commercially, with proper attribution. For more information, see our copyright guidelines: http://journals.plos.org/plosone/s/licenses-and-copyright.

4.1.         You may seek permission from the original copyright holder of Figures 4, 5, 6 to publish the content specifically under the CC BY 4.0 license.

4.2.    If you are unable to obtain permission from the original copyright holder to publish these figures under the CC BY 4.0 license or if the copyright holder’s requirements are incompatible with the CC BY 4.0 license, please either i) remove the figure or ii) supply a replacement figure that complies with the CC BY 4.0 license. Please check copyright information on all replacement figures and update the figure caption with source information. If applicable, please specify in the figure caption text when a figure is similar but not identical to the original image and is therefore for illustrative purposes only.

Reviewers' comments:

Reviewer's Responses to Questions

**Comments to the Author**

1. Is the manuscript technically sound, and do the data support the conclusions?

Reviewer #1: Yes

Reviewer #2: Yes

2. Has the statistical analysis been performed appropriately and rigorously? 

Reviewer #1: Yes

Reviewer #2: Yes

3. Have the authors made all data underlying the findings in their manuscript fully available?

Reviewer #1: Yes

Reviewer #2: Yes

4. Is the manuscript presented in an intelligible fashion and written in standard English?

Reviewer #1: Yes

Reviewer #2: Yes

5. Review Comments to the Author

Reviewer #1: In this article, the authors modeled a representative primary school building and suggested seismic retrofitting measures. The topic is current and relevant/important. The article is well organized and written. However, some additional comments are given in order to improve the article.

This statement should be referenced (page 3, lines 58-62):

“It is possible to reduce the seismic risk by improving the prevention studies and emergency plans. In this sense, it is important to analyse the seismic vulnerability of the buildings given that a large part of the losses is due to the deficient seismic behaviour of the building structures. Therefore, the study of seismic retrofitting techniques, which improves the buildings’ seismic behaviour, is mandatory.”

Possible literature where this statement could be find in:

Hadzima-Nyarko, M.; Mišetić, V.; Morić, D. Seismic vulnerability assessment of an old historical masonry building in Osijek, Croatia, using Damage Index. Journal of Cultural Heritage. 28 (2017), 140-150.

This part from the State of the Art is more suitable for Introduction:

“For example, several seismic vulnerability and risk research works have been performed on an urban scale in big cities such as Barcelona [10][11] or Lisbon [12]. The methodology used in these works is similar to the one applied in this study. The seismic behaviour of buildings in Barcelona (Spain) [10][11] was analysed via the capacity spectrum.”

Also, you stated that this methodology is similar – could you please explain differences and highlight what is the contribution of the provided research?

It is imported to highlight the novelty of the research since it has not been highlighted enough in the Introduction.

The following sentence should be more precise (page6, line 125): “School buildings have a high seismic vulnerability due to several aspects.” Please, specify which schools?

This statement should be referenced (page 7, line 166):

“There is a large number of studies about seismic retrofitting techniques applied to theoretical models and laboratory analysis.”

Possible literature where this statement could be find in:

Hadzima-Nyarko, M.; Ademović, N.; Pavić, G.; Kalman Šipoš, T. Strengthening techniques for masonry structures of cultural heritage according to recent Croatian provisions. Earthquakes and Structures, Vol. 15, No. 5 (2018) 473-485.

The part regarding the modeling of the plastic hinges (from pages 10 and 11 (lines 252-258)) area is a little bit unclear. You specified ASCE and EC8 and reference [38]. What kind of plastic hinges are implemented in the models?

It should be explained better what the authors mean when they say “Specific software”? (it is mentioned also in the figure caption and it is a little bit confusing).

It is suggested that the authors present real pictures of primary schools under consideration when describing the building.

It is suggested that the authors present the period of vibration of modeled building before and after retrofitting.

Reviewer #2: Thank you very much for inviting me to review the manuscript "Specific seismic retrofitting of a compact reinforced concrete building with X-bracings and steel jackets. Application to a primary school in Huelva”

The paper is properly written and the structure is clear.

There are many papers related to the issue of retrofitting and, specifically, retrofitting schools. However, it is interesting that the authors try to obtain the most effective retrofitting scheme considering structural, architectural and constructive parameters. In that sense, I suggest the authors to clearly state that, that is the main novelty of the manuscript. However, I have two major concerns.

Query #1. The authors use two retrofitting techniques: X-bracings and steel jackets. However, there are many other retrofitting solutions in the regulations such as the ATC-40, the FEMA 356 and the EC-08. Why have the authors just selected these two techniques among all the possibilities? This should be clearly justified in the text.

Query #2. In the manuscript, it is explained that large earthquakes (Mw≥6) of long return periods affect this region. In that sense, it would be interesting to compare the response spectrum that they use to obtain the performance point (based on the EC-08 and the Spanish maps) with that of a deterministic approach, this is, using attenuation laws from previous historic earthquakes.

6. PLOS authors have the option to publish the peer review history of their article (what does this mean?). If published, this will include your full peer review and any attached files.

Reviewer #1: No

Reviewer #2: No

---

## [Author Response · Author response to Decision Letter 0]

9 Aug 2020

Reviewer 1 Comments

Query 1: “This statement should be referenced (page 3, lines 58-62):

“It is possible to reduce the seismic risk by improving the prevention studies and emergency plans. In this sense, it is important to analyse the seismic vulnerability of the buildings given that a large part of the losses is due to the deficient seismic behaviour of the building structures. Therefore, the study of seismic retrofitting techniques, which improves the buildings’ seismic behaviour, is mandatory.”

Possible literature where this statement could be find in:

Hadzima-Nyarko, M.; Mišetić, V.; Morić, D. Seismic vulnerability assessment of an old historical masonry building in Osijek, Croatia, using Damage Index. Journal of Cultural Heritage. 28(2017), 140-150.”

Thank you for the reference. It has been added to the manuscript.

Query 2: “This part from the State of the Art is more suitable for Introduction:

“For example, several seismic vulnerability and risk research works have been performed on an urban scale in big cities such as Barcelona [10][11] or Lisbon [12]. The methodology used in these works is similar to the one applied in this study. The seismic behaviour of buildings in Barcelona (Spain) [10][11] was analysed via the capacity spectrum.”

You stated that this methodology is similar – could you please explain differences and highlight what is the contribution of the provided research?

It is imported to highlight the novelty of the research since it has not been highlighted enough in the Introduction.”

On the one hand, we completely agree with you that the methodological reference should not be included in the state of the Art section. Thus, that part of the sentence has been deleted. However, we consider that the rest of the aforementioned paragraph should be kept in the State of the Art section, since it presents other works related to the investigation. On the other hand, the main novelty of the manuscript has been properly highlighted in the introduction section: “It is important to highlight that this research try to obtain the most effective retrofitting scheme considering structural, architectural and constructive parameters”

Query 3: “The following sentence should be more precise (page 6, line 125): “School buildings have a high seismic vulnerability due to several aspects.” Please, specify which schools?”

The first draft of the manuscript included some reasons that justify the statement. However, and following your requirement, the paragraph has been modified, being more precise now.

“School buildings located in the Huelva region have a high seismic vulnerability due to the following aspects: their low adult/child ratio and their configuration. The child population (kids between 3 to 11 years old) are the most vulnerable people in the society. The building´s configuration is characterized by the presence of several seismic weak points (soft storeys at ground floors, plan irregularities or short columns).”

Query 4:This statement should be referenced (page 7, line 166):

“There is a large number of studies about seismic retrofitting techniques applied to theoretical models and laboratory analysis.”

Possible literature where this statement could be find in:

Hadzima-Nyarko, M.; Ademović, N.; Pavić, G.; Kalman Šipoš, T. Strengthening techniques for masonry structures of cultural heritage according to recent Croatian provisions. Earthquakes and Structures, Vol. 15, No. 5 (2018) 473-485.

Thank you for the reference. It has been added to the manuscript.

Query 5: “The part regarding the modeling of the plastic hinges (from pages 10 and 11 (lines 252-258)) area is a little bit unclear. You specified ASCE and EC8 and reference [38]. What kind of plastic hinges are implemented in the models?”

Plastic Hinges have been defined according to ASCE 41—13 failure criteria. Their locations on the structural elements have been defined according to the recommendations given by EC-08.

 “The nonlinear behaviour of RC elements has been simulated by adding plastic hinges as in [40]. The hinges have been defined according to ASCE-41-13 [39] failure criteria. Regarding their location, the hinges have been added at the ends of both beams and columns (5% and 95% of the total element length), as recommended in the EC-8.”

Query 6:“It should be explained better what the authors mean when they say “Specific software”? (it is mentioned also in the figure caption and it is a little bit confusing).”

We agree with this statement. It has been explained and referenced in the paper.

“the specific software, which has been developed in the PERSISTAH project to evaluate the seismic vulnerability of building´s schools [49].”

Query 7: “It is suggested that the authors present real pictures of primary schools under consideration when describing the building.”

Thank you for your suggestion. Real pictures of the school’s typology under study have been included.

 “Fig 7. Pictures of the school´s typology selected as case study (Author’s ownership)”

“It is suggested that the authors present the period of vibration of modelled building before and after retrofitting.”

The period of vibration of modelled building, before and after retrofitting, have been added.

“The main period of vibration of the original building was found to be T=1.022s in X direction. After the retrofitting (SJ3), this period shifted to T=0.906s. Regarding the Y direction, the original period was of T=0.84s, decreasing down to T=0.73s after retrofitting (XB-Y 2).”

Reviewer 2 Comments

Query 1: “There are many papers related to the issue of retrofitting and, specifically, retrofitting schools. However, it is interesting that the authors try to obtain the most effective retrofitting scheme considering structural, architectural and constructive parameters. In that sense, I suggest the authors to clearly state that, that is the main novelty of the manuscript.”

Thank you for your kind comments. The main novelty of the manuscript has been properly highlighted in the document.

Query 2:“The authors use two retrofitting techniques: X-bracings and steel jackets. However, there are many other retrofitting solutions in the regulations such as the ATC-40, the FEMA 356 and the EC-08. Why have the authors just selected these two techniques among all the possibilities? This should be clearly justified in the text.”

The justification of the use of these retrofitting techniques has been incorporated in the text.

“Based on the analysis of the weak points of the building, two seismic retrofitting techniques have been selected to study their effectiveness, (i) Steel X-Bracing, and (ii) Steel Jackets (Fig 6). The selection of these retrofitting techniques has been selected considering the following aspects: constructability, architectural impact (both aesthetics and functional), cost and hindrance to educational activity.”

Query 3: “In the manuscript, it is explained that large earthquakes (Mw≥6) of long return periods affect this region. In that sense, it would be interesting to compare the response spectrum that they use to obtain the performance point (based on the EC-08 and the Spanish maps) with that of a deterministic approach, this is, using attenuation laws from previous historic earthquakes.”

A Figure with the comparison of both response spectra has been incorporated in the manuscript.

“Fig 1. Comparison of response spectrums of EC-8 and NCSE-02 with the response spectrums of historic earthquakes.”

Academic Editor Comments

“We note that Figures 4, 5, 6 in your submission contain copyrighted images.”

Dear Editor,

Figures 4 and 6 (now Fig. 5 and 8) have been made by the authors, based on the original blueprints of the school obtained from the local file. A copy of the original file has been uploaded as other item. Figure 5 (now 6) is a completely new figure made by the authors.

---

## [Decision Letter · Decision Letter 1]

19 Aug 2020

Specific seismic retrofitting of a compact reinforced concrete building with X-bracings and steel jackets. Application to a primary school in Huelva.

PONE-D-20-15119R1

Dear Dr. Morales-Esteban,

We’re pleased to inform you that your manuscript has been judged scientifically suitable for publication and will be formally accepted for publication once it meets all outstanding technical requirements.

Kind regards,

Francisco Martínez-Álvarez

Academic Editor

PLOS ONE

Additional Editor Comments (optional):

Reviewers' comments:

Reviewer's Responses to Questions

**Comments to the Author**

1. If the authors have adequately addressed your comments raised in a previous round of review and you feel that this manuscript is now acceptable for publication, you may indicate that here to bypass the “Comments to the Author” section, enter your conflict of interest statement in the “Confidential to Editor” section, and submit your "Accept" recommendation.

Reviewer #1: All comments have been addressed

Reviewer #2: All comments have been addressed

2. Is the manuscript technically sound, and do the data support the conclusions?

Reviewer #1: Yes

Reviewer #2: Yes

3. Has the statistical analysis been performed appropriately and rigorously? 

Reviewer #1: Yes

Reviewer #2: Yes

4. Have the authors made all data underlying the findings in their manuscript fully available?

Reviewer #1: Yes

Reviewer #2: Yes

5. Is the manuscript presented in an intelligible fashion and written in standard English?

Reviewer #1: Yes

Reviewer #2: Yes

6. Review Comments to the Author

Reviewer #1: Thank you for making corrections according to all recommendations and thus improving the manuscript. The article can be published in present form.

Reviewer #2: The manuscript has been corrected according to the reviewer's suggestions, and for this reason and from my point of view, it should be accepted to be published in the journal "plos one"

7. PLOS authors have the option to publish the peer review history of their article (what does this mean?). If published, this will include your full peer review and any attached files.

Reviewer #1: No

Reviewer #2: No

---

## [Editor Report · Acceptance letter]

1 Sep 2020

PONE-D-20-15119R1 

Specific seismic retrofitting of a compact reinforced concrete building with X-bracings and steel jackets. Application to a primary school in Huelva. 

Dear Dr. Morales-Esteban:

I'm pleased to inform you that your manuscript has been deemed suitable for publication in PLOS ONE. Congratulations! Your manuscript is now with our production department. 

Kind regards, 

on behalf of

Dr. Francisco Martínez-Álvarez 

Academic Editor

PLOS ONE